# Fibrinolytic-deficiencies predispose hosts to septicemia from a catheter-associated UTI

Jonathan J. Molina[1,2,11], Kurt N. Kohler [2,11], Christopher Gager[2], Marissa J. Andersen[2], Ellsa Wongso[2], Elizabeth R. Lucas [2], Andrew Paik [2], Wei Xu [3,4], Deborah L. Donahue[5,6], Karla Bergeron[7], Aleksandra Klim[7], Michael G. Caparon[3,4], Scott J. Hultgren [3,4], Alana Desai[7,8], Victoria A. Ploplis[5,6], Matthew J. Flick[9,10], Francis J. Castellino [5,6] & Ana L. Flores-Mireles [1,2,5] ✉

Catheter-associated urinary tract infections (CAUTIs) are amongst the most common nosocomial infections worldwide and are difficult to treat partly due to development of multidrug-resistance from CAUTI-related pathogens. Importantly, CAUTI often leads to secondary bloodstream infections and death. A major challenge is to predict when patients will develop CAUTIs and which populations are at-risk for bloodstream infections. Catheter-induced inflammation promotes fibrinogen (Fg) and fibrin accumulation in the bladder which are exploited as a biofilm formation platform by CAUTI pathogens. Using our established mouse model of CAUTI, here we identified that host populations exhibiting either genetic or acquired fibrinolytic-deficiencies, inducing fibrin deposition in the catheterized bladder, are predisposed to severe CAUTI and septicemia by diverse uropathogens in mono- and poly-microbial infections. Furthermore, here we found that *Enterococcus faecalis*, a prevalent CAUTI pathogen, uses the secreted protease, SprE, to induce fibrin accumulation and create a niche ideal for growth, biofilm formation, and persistence during CAUTI.

Urinary catheterization is a common procedure to drain urine from patients' bladders due to chronic conditions or while in healthcare facilities, intensive care units, during surgical procedures and recovery[1–3]. Despite its benefits, catheter placement increases the risk of developing a catheter-associated urinary tract infection (CAUTI)[4–7]. CAUTIs are one of the most common nosocomial infections and often lead to septicemia with a 30% mortality[3,8]. In fact, ~25% of sepsis cases come from complicated UTI, including CAUTIs[9]. Current CAUTI management focuses on catheter removal, replacement, and an antibiotic regimen[6]. However, the consistent colonization of catheters by external and fecal microflora, along with polymicrobial infections, pose challenges to management and treatment due to the increasing prevalence of antibiotic-resistant CAUTI pathogens[6,10,11]. Leading the CDC and WHO to classify CAUTI as a serious threat[12,13].

Currently, there is no consensus on best practices for CAUTI treatment. Patients are treated with the same protocols as non-catheter associated (uncomplicated) urinary tract infections (uUTI). However, CAUTIs exhibit unique clinical manifestations, causative

[1]Integrated Biomedical Sciences, University of Notre Dame, Notre Dame, IN 46556, USA. [2]Department of Biological Sciences, University of Notre Dame, Notre Dame, IN 46556, USA. [3]Department of Molecular Microbiology, Washington University School of Medicine, St. Louis, MO 63110, USA. [4]Center for Women's Infectious Disease Research, Washington University School of Medicine, St. Louis, MO 63110, USA. [5]W. M. Keck Center for Transgene Research, University of Notre Dame, Notre Dame, IN 46556, USA. [6]Department of Chemistry and Biochemistry, University of Notre Dame, Notre Dame, IN 46556, USA. [7]Department of Surgery, Washington University School of Medicine, St. Louis, MO 63110, USA. [8]Department of Urology, University of Washington Medical Center, Seattle, WA 98133-9733, USA. [9]Department of Pathology and Laboratory Medicine, University of North Carolina, Chapel Hill, NC 27599, USA. [10]UNC Blood Research Center, University of North Carolina, Chapel Hill, NC 27599, USA. [11]These authors contributed equally: Jonathan J. Molina, Kurt N. Kohler. ✉ e-mail: afloresm@nd.edu

organisms, and pathologic mechanisms, making these infections distinct from uUTI[6]. For example, uUTIs are more prevalent in women than men (4:1 ratio) while in CAUTI there is no gender bias[6]. Also, *E. coli* accounts for >95% of uUTI, whereas CAUTI-pathogens are more diverse, including gram-negative, gram-positive, and fungal pathogens[6].

In both humans and mice, urinary catheterization provokes local tissue damage to the bladder, activating coagulation, and converting the blood coagulation protein, fibrinogen (Fg), into fibrin. Fibrin clots and extravascular fibrin deposits promote blood clotting and wound healing[14]. Then, in fibrinolysis, plasminogen (Pg) is activated into serine protease plasmin to degrade fibrin clots and restore tissue homeostasis[14,15]. However, constant catheter-induced bladder inflammation induces Fg/fibrin accumulation on the catheter with increasing catheterization time in human and mice[16,17]. Fg/fibrin deposition compromises the urothelium, thereby introducing a platform for biofilm formation by CAUTI pathogens[16–23].

Here, we identified that mice with genetic or acquired fibrinolytic-deficiencies are susceptible to severe and persistent CAUTI and systemic dissemination by diverse groups of prevalent uropathogens. Moreover, we found that *E. faecalis*, a prevalent CAUTI pathogen known to bind to Fg, secretes a protease, SprE, to induce fibrin accumulation and create a niche ideal for growth, biofilm formation and persistence during CAUTI. Furthermore, antifibrinolytic agent usage in catheterized mice results in Fg/fibrin accumulation in the bladders, resulting in persistent CAUTI and systemic dissemination in mono- and poly-microbial infection by *E. faecalis*, *E. coli*, and *C. albicans*. Identification of patient populations with higher susceptibility to CAUTI and its sequelae will allow clinicians to improve patient outcomes by implementing efficient patient monitoring to mitigate infection incidences, morbidities, and mortality.

## Results

### Prolonged urinary catheterization promotes enterococcal burden and persistence in the bladder and systemic dissemination

Clinical studies showed that CAUTI-burdened patients have an increased risk of septicemia and mortality[10,11,24–26]. To understand *E. faecalis* bladder and catheter colonization progression and systemic dissemination, we performed a temporal study examining acute infection (1, 3, 6, 9, 12 hr post-catheterization and infection (hpi) and 1 day post-infection (dpi)) and prolonged infection (3, 7, 14 dpi). Female C57BL/6 mice were catheterized and infected with *E. faecalis* (-2 ×10^7 CFU) or mock-infected (PBS). At specified timepoints, mice were sacrificed to harvest the bladders, catheters, kidneys, spleens, and hearts. Bladders at 1 hpi have an initial colonization of -10^4 bacterial CFU that increased significantly overtime, showing maximum colonization at -10^7 CFU by 12 hpi in the bladder and persisting through 14 dpi (Fig. 1a). Similarly, catheter colonization significantly increased with catheterization time and persisted (Fig. 1b). Importantly, robust bladder and catheter colonization allowed *E. faecalis* to disseminate to the kidneys, spleen, and heart (Fig. 1c–e). Thus, our data recapitulates what is clinically observed[11,12,25–27].

### Progression of bladder inflammation correlates with microbial burden

Catheter-induced inflammation is a CAUTI hallmark[21,27,28]. To understand how urinary catheterization changes the bladder environment during acute and prolonged catheterization and in the presence or absence of infection; we examined mouse bladder edema and inflammation by bladder weight and histological analysis. Bladders exhibited a progressive weight increase with the catheter dwell time (Fig. 1f). Bladder histological analysis corroborates the gradual edema progression and increase of the bladder size associated with urinary catheterization (Fig. 1g). Furthermore, there is a significant positive correlation of bladder weight with bacterial burden in the bladder

(r = 0.7991; P = 0.0098) or catheter (r = 0.9005; P = 0.0008) (Supplementary Fig. 1). Importantly, *E. faecalis* infection further increased bladder weight after 12 hpi (Fig. 1f). This indicates that catheterization alone increases bladder edema overtime and is exacerbated by an *E. faecalis* infection.

### *E. faecalis* presence modulates levels of inflammatory cytokines in the catheterized bladder

Previous studies show that acute urinary catheterization induces many inflammatory cytokines in the bladder[27], of which IL-1β, IL-6, 12(p40), IL-17, CSF 3, and CXCL1 are further induced during *E. faecalis* infection[28]. Since *E. faecalis* significantly exacerbates bladder inflammation during catheterization (Fig. 1f, g), we profiled cytokine level changes of *E. faecalis*- or mock-infected catheterized bladder, relative to naïve bladders. We found that *E. faecalis* infection altered cytokine expression patterns differently than in mock-infected bladders. IL-6, CSF 3, CXCL1, and IL-17 levels were significantly higher during infection (Fig. 1h, i, and Supplementary Fig. 2). Conversely, infection significantly reduced levels of IL-1α, IL-2, IL-9, IL-10, IL-12 (p70), IL-13, IFNγ, IL-3, CCL4 (MIP-1β), CSF 2 (GM-CSF), CCL5 (RANTES), and TNFα (Fig. 1h, Supplementary Figs. 2 and 3a, c). IL-1β, IL-12 (p40), IL-4, and CCL3 levels were modulated, showing significantly lower induction during early infection (1–9 h) followed by a significant increased after 12 h (Fig. 1h, Supplementary Figs. 2 and 3b). IL-5 and CCL2 levels increased regardless of the infection (Fig. 1h and Supplementary Fig. 2). This indicates that there is a differential inflammatory cytokine profile on infected catheterized bladders.

During catheter-induced bladder inflammation in mice and humans[16,17], Fg, a liver-produced proinflammatory protein, is recruited into the bladder and deposited on urinary catheters, serving as a platform for microbial CAUTI[16,18,21–24,29]. IL-1 (α and β), IL-6, and TNFα are known to stimulate Fg production by migrating from damaged tissues, through the bloodstream and signaling to Fg-producing hepatocytes in the liver[30,31]. To understand their contribution to CAUTI-associated inflammation, we evaluated levels of these proinflammatory mediators in the bladder. High levels of IL-6 during catheterization were observed, which significantly increased during infection (Fig. 1h, i). We also assessed the levels of all four cytokines in the bloodstream and livers. Expectedly, IL-6 levels were elevated in the bloodstream and liver (Fig. 1i and Supplementary Fig. 3). To understand the role of IL-1α, IL-1β, IL-6 and TNFα in human urinary catheterization, we collected urine from patients that had a urinary catheter for -24 h and from healthy donors to compare and analyze their levels. We found that IL-1α, IL-1β, IL-6 and TNFα levels significantly increased in urine from catheterized patients when compared with urine from non-catheterized healthy donors (Fig. 1j). Furthermore, IL-6 was significantly elevated compared with IL-1α, IL-1β, and TNFα in the catheterized patients (Fig. 1j). A limitation of this study is that urine samples were not collected from catheterized patients before catheterization as a further pair-wise comparison. Without this data, any comorbidity prior to catheterization could also contribute to the cytokines present in patient urine post-catheterization. However, based on this data provided as is, we can speculate IL-6 could be an important cytokine during human and mouse CAUTIs and, in mice, IL-6 may communicate with the liver's Fg expression and release, resulting in its accumulation in the catheterized bladder. Further work should be done to fully assess this.

### Fg deposition increases on urinary catheters during *E. faecalis* infection

Previously, clinical and animal studies from our and other groups have shown that Fg is a major component of proteins deposited on urinary catheters[16,17,19–21,23,32,33]. Importantly, our previous immunofluorescence analyses on catheters showed increased Fg deposition during *E. faecalis* infection[16–18], suggesting that infection may

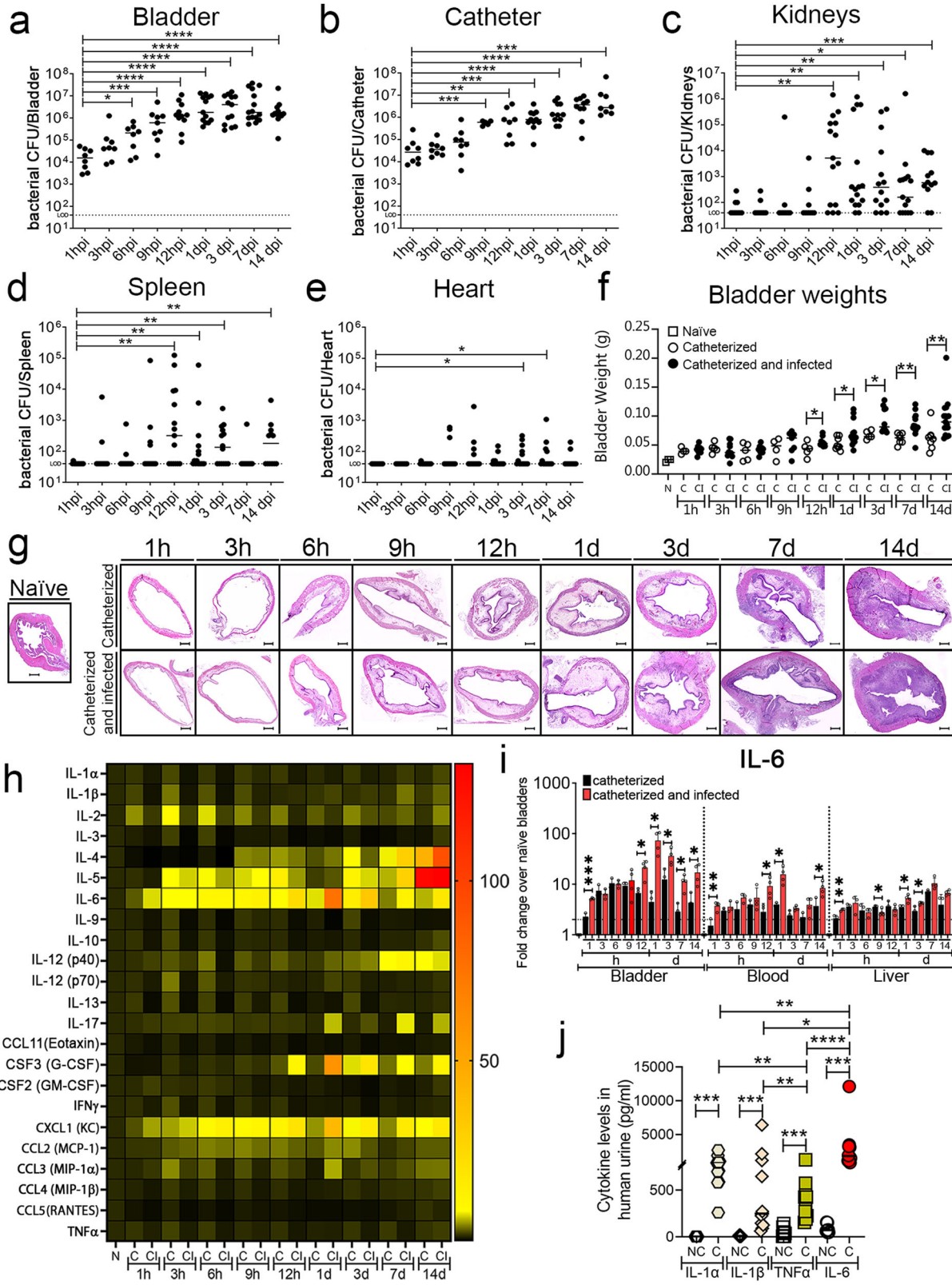

promote Fg recruitment. To test this, we performed a temporal quantitative proteomic analysis to compare Fg abundance on urinary catheters by analyzing the three Fg chains, α, β, γ, between catheterized (mock-infected) and catheterized and infected mice. Five catheters were harvested at the indicated timepoints, pooled, processed, and trypsin digested. Total protein analysis showed 230 proteins associated with mouse catheters (Fig. 2a, c, and

Supplementary Data 1). Fg is among the most abundant deposited proteins on the catheter, corroborating our previous proteomic study at 24 hpi[19] and its abundance significantly increased in acute and prolonged infection starting 1 dpi (Fig. 2b, Supplementary Fig. 4, and Supplementary Data 1). Higher Fg -α, -β, -γ total peptide counts in infected bladders suggest that *E. faecalis* may have a mechanism to promote Fg accumulation.

**Fig. 1 | *E. faecalis* infection and inflammation progression during urinary catheterization. a–e** *E. faecalis* establishes persistent colonization overtime. Mice were catheterized and infected with $2 \times 10^7$ CFU of *E. faecalis* OG1RF. Enterococcal colonization of organs and catheters were assessed by quantifying bacterial burden. **f** Bladder weights of naïve (non-implanted control) mice and implanted mice in the presence or absence of the *E. faecalis* at the indicated times. **g** Bladder sections were stained with H&E to compare inflammation from catheterization in the presence or absence of the pathogen overtime (scale bars: 500 μm). **h** Heatmap represents catheterized bladder cytokines fold changes over naïve bladder in the presence or absence of the *E. faecalis* OG1RF infection at the indicated times. **i** IL-6 analysis levels in the catheterized bladder (with or without infection), bloodstream, and liver. The horizontal broken line represents twofold change cytokine levels compared with naïve control mice over a 2-fold change with the bar representing the mean ± SD. **j** Comparison of the IL-1α (octagons), IL-1β (diamonds), IL-6 (circle), and TNFα (square) levels in urine from patients catheterized for 24 h and healthy donors (NC; non-catheterized). Two-tailed Mann–Whitney U test was used to determine significance; *$P < 0.05$ was considered statistically significant. **$P < 0.005$; ***$P < 0.0005$; ****$P < 0.0001$. The horizontal bar represents the median value with range as error bars. The horizontal broken line represents the limit of detection of viable bacteria. LOD limit of detection. Infections were done in three independent experiments with $n = 6$ mice for each one, and data are shown as the bacterial CFU/organ or catheter. Animals that lost the catheter were not included in this work. All $n$ values, exact $p$-values, and source data are provided in Source Data file.

## Coagulation and immune response pathway proteins are found on human and mouse urinary catheters

To understand if similar host proteins are deposited in human urinary catheter, we performed a proteomic analysis of 10 human urinary catheters with a 24–26 h dwell time, finding a total of 104 host deposited proteins, including several coagulation factors, including Fg, alpha-1 antitrypsin (A1AT), alpha-2-macroglobulin (A2M), plasminogen (Pg), antithrombin III (AT3), thrombin (THB), and kininogen (KNG) (Fig. 2a, Supplementary Data 2). Resembling mouse catheters, Fg is one of the most abundantly deposited protein. Comparative analyses showed 76 (73% of the identified human proteins) shared deposited proteins between mouse and human catheters (Fig. 2a). The 76 shared proteins were subjected to an unbiased and comprehensive pathway enrichment analyses using Metascape[34], finding that the top significant pathways enriched are part of host inflammation, immune, hemostasis, and wound healing responses, including platelet degranulation, neutrophil degranulation, complement and coagulation cascades (Fig. 2a, c, Supplementary Fig. 5a, and Supplementary Table 1). Furthermore, we found that transcriptional regulators involved in inflammation and immune responses, including CEBPA, STAT3, and NFKB1, were highly predicted in the catheterized bladder environment (Supplementary Fig. 5b and Supplementary Table 2). Additionally, we performed network analysis of differential proteins found ether in mouse or patient catheters (Supplementary Fig. 6), finding that some of the proteins are also involved in immune and hemostasis processes described in the shared proteins (Fig. 2). Collectively, proteomics and ontology relationships indicate that coagulation and wound healing pathways are active in mouse and human catheterized bladders.

## Fibrinogen is critical for colonization of the catheterized bladder and catheter

To investigate if Fg is critical for enterococcal CAUTI establishment, we compared bacterial colonization in C57BL/6 wild-type (WT) mice to several mutant mice (C57BL/6-background), including: i) Fg deficient mice (Fg$^{-/-}$)[35]; ii) mice expressing a mutant form of Fg that cannot be converted to fibrin by thrombin, remaining a monomer (Fg$^{AEK}$)[36]; and iii) FVIItTA/tTA (FVII$_{tTA}$)[37] mice, which are hypomorphic for coagulation factor VII (FVII)[38], resulting in severe thrombin production downregulation, reducing fibrin formation (Fig. 3a). Mice were catheterized and infected; then at 1 dpi, bacterial burden was assessed in organs and catheters. Compared to WT mice, Fg$^{-/-}$ mice exhibited significantly reduced enterococcal burdens in bladders (-1.5 logs) and catheters (-3.5 logs) (Fig. 3b, c). Similarly, Fg$^{AEK}$ mice exhibited significant defects in bladder and catheter colonization (Fig. 3b, c). Fibrin reduction in the bladder via FVII$^{tTA}$, resulted in significantly lower colonization in the bladders (-1 log) and catheters (-1.5 logs) (Fig. 3b, c). Soluble Fg presence (Fg$^{AEK}$) significantly decreased kidney dissemination (Fig. 3d). Enterococcal dissemination to the spleen and hearts was not observed in any of the mouse lines (Fig. 3e, f). These results further confirm that Fg is critical for enterococcal CAUTI. Importantly, soluble Fg or low fibrin formation also results in lower colonization, indicating that fibrin accumulation is critical for persistent colonization.

## Fibrin accumulation enhances enterococcal bladder and catheter persistence during CAUTI

Fibrinolysis is critical for dissolving fibrin net/clots and defective fibrinolysis results in fibrin accumulation. Based on this and that Pg was found on the human and mouse catheters (Fig. 2a), we investigated fibrinolysis in CAUTI, focusing in Pg and its key activators. We used C57BL/6-background mice deficient in different fibrinolysis factors (Fig. 3a). To increase fibrin accumulation, we used mouse lines deficient for: i) plasminogen (Pg$^{-/-}$)[39], ii) urokinase plasminogen activator (uPA$^{-/-}$)[40], and iii) tissue plasminogen activator (tPA$^{-/-}$)[40]. To decrease fibrin levels, we used a deletion of plasminogen activator inhibitor (PAI$^{-/-}$)[41], which exhibits unregulated fibrin degradation (Fig. 3a).

We found that persistent fibrin accumulation led to significant increases of enterococcal bladder (-2 logs) and catheter (-1.5 logs) colonization in Pg$^{-/-}$ mice when compared to WT mice (Fig. 3g, h). Importantly, fibrin reduction in the bladder via PAI deficiency, resulted in significantly lower bladder (-1 log) and catheter (-1.5 logs) colonization (Fig. 3g, h).

To further confirm that inactivation of Pg proteolytic activity correlates with enterococcal colonization enhancement, we used a transgenic mouse expressing Pg with a plasmin-inactivating active site mutation (PGB)[42], finding that bacterial burden in the bladder and catheter was similar to Pg$^{-/-}$ mice (Fig. 3g, h). This suggests that fibrinolysis inhibition correlates with exacerbating *E. faecalis* colonization.

Importantly, bladder and catheter bacterial colonization in uPA$^{-/-}$ mice significantly increased similarly to Pg$^{-/-}$ mice, whereas tPA$^{-/-}$ mice exhibit similar bacterial burden to WT mice. Thus, uPA is the primary activator of Pg into plasmin in the catheterized bladder (Fig. 3g, h). Since coagulation dysregulation is not uncommon in human populations (Supplementary Table 3), these data suggest that coagulation deregulation could have a direct impact on enterococcal CAUTI.

## Host fibrinolytic deficiencies predispose the host to enterococcal systemic dissemination

We further assessed the effect of Fg/fibrin modulation in enterococcal dissemination, finding that fibrinolytic defects (Pg$^{-/-}$, PGB, and uPA$^{-/-}$) significantly increased bacterial burden in kidneys, spleen, and hearts compared with WT mice colonization (Fig. 3i–k). Furthermore, tPA deficiency resulted in a bimodal kidney enterococcal colonization (Fig. 3i). Interestingly, kidneys with higher enterococcal colonization came from mice with higher bladder colonization, suggesting that a persistent bladder colonization may result in their dissemination (Fig. 3i). These demonstrate that fibrin accumulation predisposes the host to enterococcal systemic dissemination.

### *E. faecalis* proteases degrade plasmin and plasminogen but not thrombin

We previously demonstrated that during CAUTI and growth in urine conditions, *E. faecalis* OG1RF induced the expression and activity of

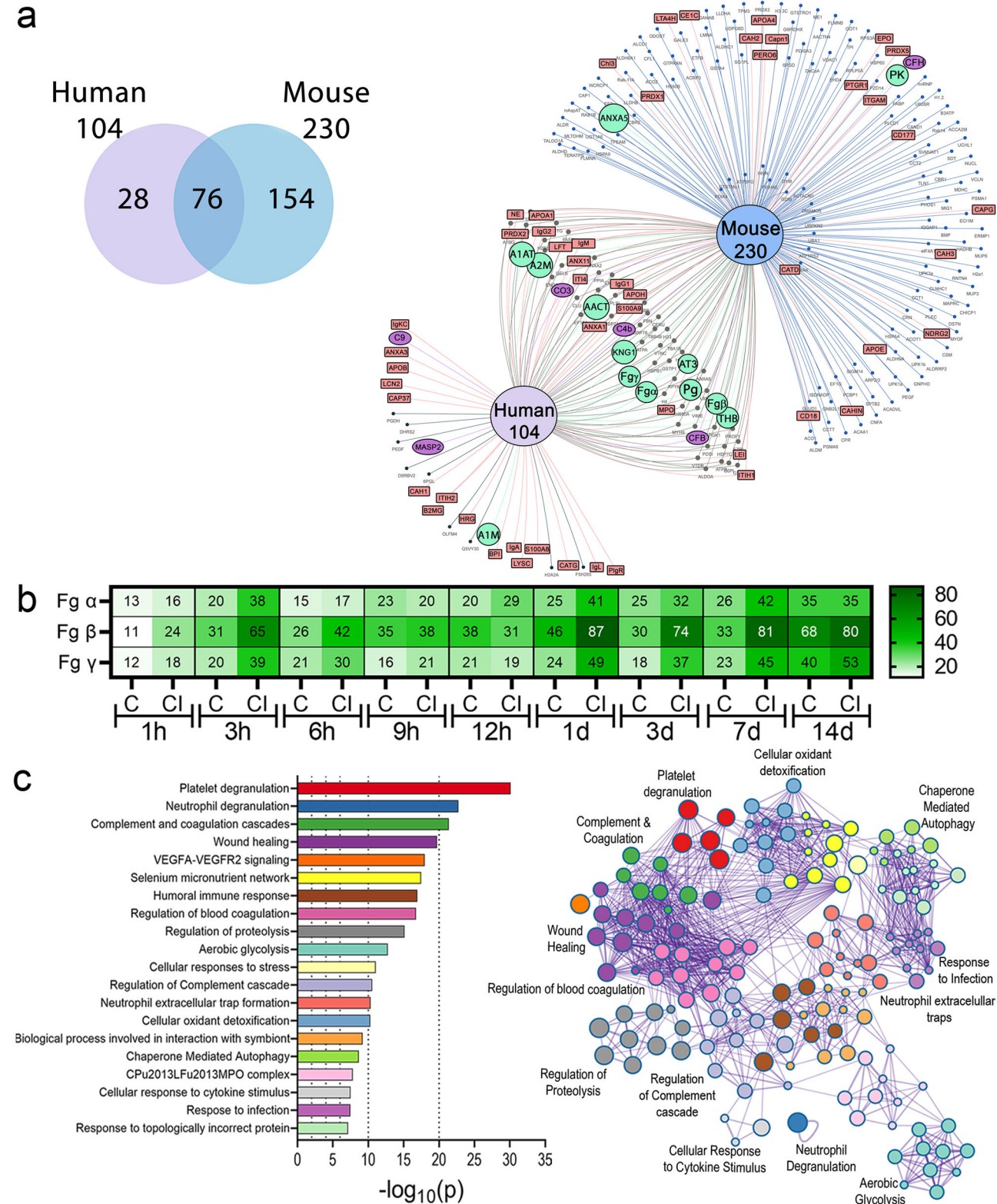

**Fig. 2 | Proteomic analysis of the proteins deposited on urinary catheters retrieved from patients and mice. a** Venn diagram comparing deposited proteins on urinary catheters retrieved from patients and mice catheterized with a dwell time of 1 day using E Venn online software[S1]. Red rectangles represent inflammatory and immune proteins, green circles represent coagulation cascade proteins, purple ovals represent complement cascade proteins, while black dots represent other proteins. **b** Fg levels in acute and prolonged catheterized bladder in the absence or presence of *E. faecalis* OG1RF infection. **c** Metascape analysis including a hypergeometric test and Benjamini−Hochberg p-value correction showing the top significant pathways shared proteins found on mice and human catheters based on corrected -log(p-value). Metascape network analysis and clusters of interactions of the top significant pathways were visualized with Cytoscape[S2]. Colors on bar graph correspond to the gene ontologies on the adjacent network analysis. Source data is provided in Source Data file.

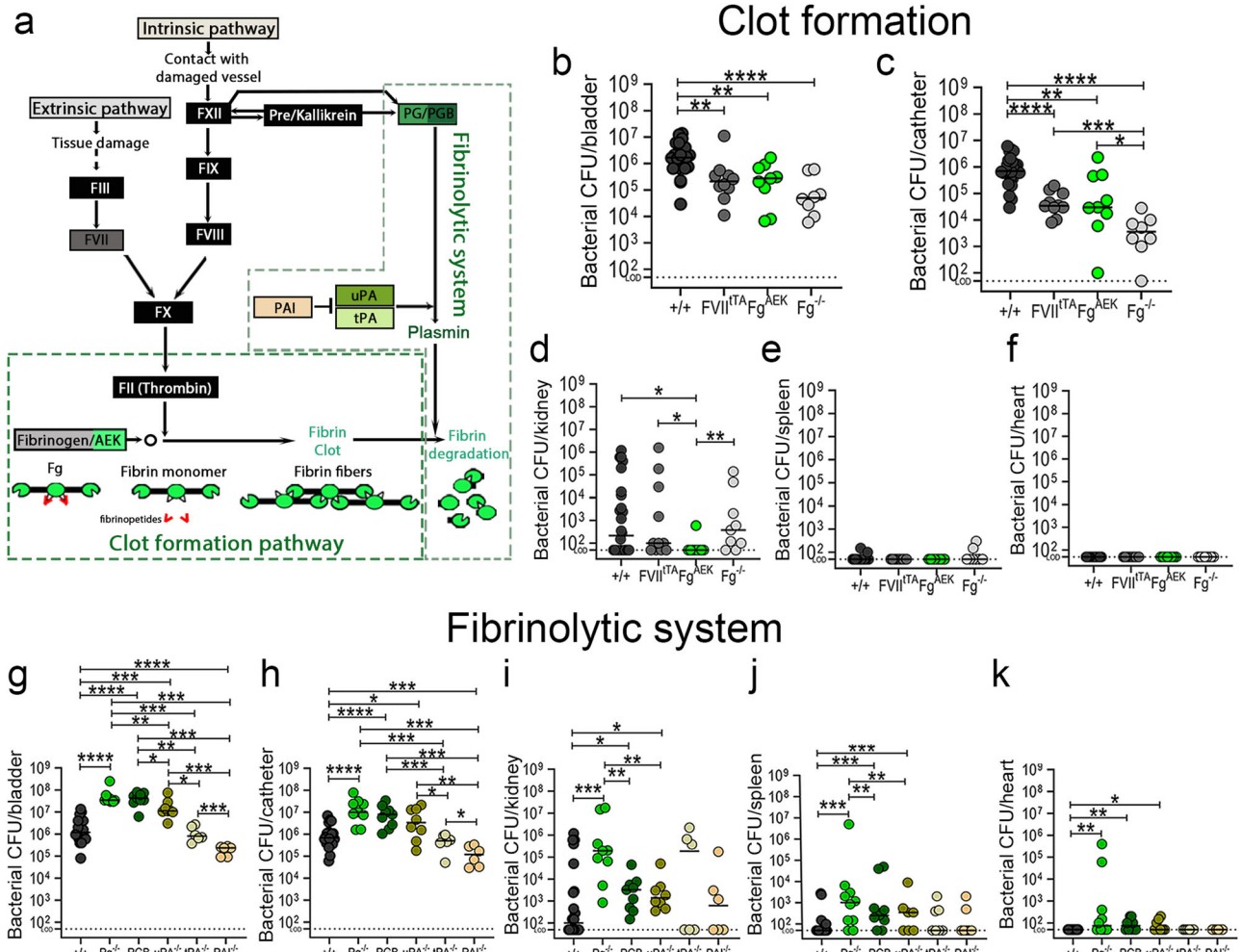

**Fig. 3 | Impairment of the fibrinolysis enhances enterococcal colonization and systemic dissemination. a** Coagulation cascade diagram (color boxes correlates with mouse strains used in this study). C57BL/6 wild type (WT) mice and transgenic coagulation mutants in C57BL/6-background looking at clot formation pathway (**b–f**) or fibrinolytic system (**g–k**) were catheterized and infected with ~2 × 10⁷ CFU of *E. faecalis* OG1RF. After 24 hpi, bacterial burdens were measured in bladder tissues (**a, g**), catheters (**b, h**), kidneys (**c, i**), spleen (**d, j**), and hearts (**f, k**). For **b–f**: *n* = 22 for +/+, *n* = 8 for Fg−/−, *n* = 10 for FVII^tTA, and *n* = 9 for Fg^AEK. For **g–k**) *n* = 19 for +/+, *n* = 9 for Pg⁻/⁻, *n* = 9 for PGB, *n* = 8 for uPA⁻/⁻, *n* = 6 for tPA⁻/⁻, and *n* = 6 for PAI⁻/⁻.

The Kruskal–Wallis test followed by a two-tailed Dunn's test was used to determine significance; *P < 0.05 was considered statistically significant. **P < 0.005; ***P < 0.0005; ****P < 0.0001. The horizontal bar represents the median value. The horizontal broken line represents the limit of detection of viable bacteria. LOD limit of detection. Colors are used to help distinguish between genotypes. For CFU enumeration, infections were done in at least three independent experiments with *n* = 3–6 mice depending on the genotype, and data are shown as the bacterial CFU/ organ or catheter. Total *n* Animals that lost the catheter were not included in this work. All *n* values, exact *p*-values, and source data are provided in Source Data file.

two enterococcal secreted proteases, a serine protease (SprE) and metalloproteinase gelatinase (GelE), which are critical for enterococcal CAUTI and systemic dissemination[29]. Given that *E. faecalis* in the catheterized bladder increases Fg/fibrin levels (Fig. 2b) and that fibrinolysis deficiencies enhance enterococcal colonization (Fig. 3), we examined if the enterococcal secreted proteases antagonizes the fibrinolytic system by assessing their ability to degrade human Pg (hPg) and plasmin (hPm) in vitro.

We compared the protease activity of WT, mutants defective for each protease (ΔsprE or ΔgelE), or both proteases (ΔgelEΔsprE) against hPg and hPm under urine conditions (See Methods, Supplementary Fig. 7). Filtered WT supernatants degraded hPg (~95 kDa) and hPm (~85 kDa), producing a major degradation fragment ~35 kDa and 3 visible degradation products (DP) between 50–40 kDa (Fig. 4a, b). WT and ΔgelE supernatants produced similar hPg and hPm degradation pattern while the ΔsprE supernatant was not able to fully degrade hPg and hPm, showing three faint fragments between 50–40 kDa and no major degradation product at ~35 kDa (Fig. 4a, b). The ΔgelEΔsprE mutant completely lost the ability to degrade hPg

and hPm into smaller peptides, similar to urine or PBS controls (Fig. 4a, b, d, e).

To understand whether these proteases are promiscuous and capable of targeting other factors in the coagulation cascade, specifically the fibrin formation pathway, we incubated the strains' supernatants with thrombin, finding that thrombin was not degraded in any of the treatments (Fig. 4c, f). To further explore SprE specificity, we tested degradation of other host proteins with comparable size to Pg and Pm, prekallikrein (~90 kDa), factor XII (FXII; ~80 kDa), and endoplasmin (~110 kDa). Prekallikrein and FXII are part of the coagulation cascade (Fig. 3a) and endoplasmin (HSP090b1; ~110 kDa observed) was found deposited on both mouse and patient catheters with an expected size of ~90 kDa, similar to Pg (Supplementary Data 1 and 2). We found differential degradation of these proteins by the enterococcal proteases (Supplementary Fig. 8), GelE was important for degradation of FXII while SprE was important for degradation for prekallikrein but dispensable for endoplasmin and FXII. This suggests that enterococcal secreted proteases could modulate the coagulation cascade and other host pathways.

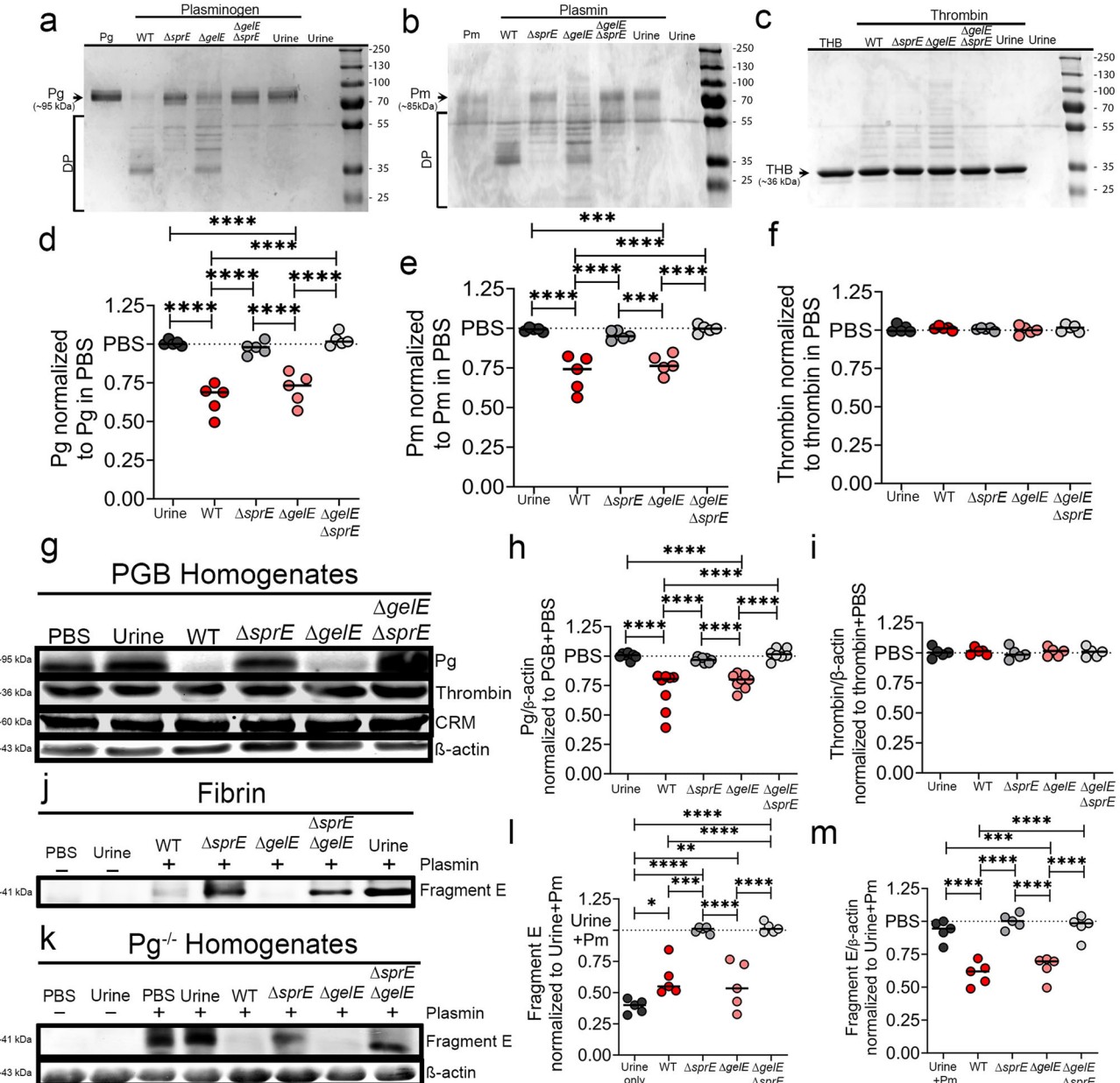

**Fig. 4 | SprE, an *E. faecalis* secreted protease, selectively degrades plasminogen and plasmin, inactivating plasmin proteolytic activity against fibrin.** SDS-PAGE analysis of the proteolytic activity of *E. faecalis* WT and protease mutants' cell-free supernatants against purified (**a**) plasminogen, (**b**) plasmin, or (**c**) thrombin and their corresponding degradation quantification by densitometry (**d**–**f**). **g** 24 h catheterized PGB bladder homogenates were incubated with *E. faecalis* WT and protease mutants' cell-free supernatants and proteolytic activity against plasminogen and thrombin was monitored by Western blots. **h, i** Densitometry analysis of the Pg and thrombin degradation by bacterial supernatants performed in **g**. **j**–**m** SprE degradation of plasmin results in inhibition of fibrinolysis. To test this, supernatants *E. faecalis* grown in urine were filtered and concentrated, then incubated plasmin was for 4 h at 37 °C; then each mixture was incubated with purified (**j**)

fibrin or (**k**) Pg⁻/⁻ mouse bladder homogenates from 24 h catheterized non-infected mice. Degradation of fibrin was monitored by detection of Fragment E in incubation with fibrin or bladder homogenates by SDS-PAGE or western blot analysis, respectively (**j, k**). Fragment E quantification by densitometry obtained in **l** and **m**. CRM, cross reactive material. β-actin was used as loading and normalization control. An ANOVA followed by a two-tailed Tukey's post-hoc was used to determine significance; *$P < 0.05$ was considered statistically significant. **$P < 0.005$; ***$P < 0.0005$. For **d**–**f**: $n = 5$; **h**, $n = 8$; **i**, $n = 5$; and **l, m**, $n = 5$. The horizontal bar represents the median value. Blot images (**g, k**) were processed in parallel and sample processing controls were run on different gels. All *n* values, exact *p*-values, and source data including uncropped gel images are provided in Source Data file.

To further validate the Pg and Pm degradation, we determined the proteolytic activity of supernatants from *E. faecalis* WT and various mutants against mouse Pg and thrombin in a complex catheterized environment. Cell-free bacterial supernatants were incubated with bladder homogenates (1:5) from PGB mice catheterized for 24 h (Supplementary Fig. 7). WT and Δ*gel*E supernatants significantly degraded Pg while no degradation was observed in the urine control, the Δ*spr*E single mutant and the double Δ*gel*E Δ*spr*E mutant

supernatant (Fig. 4g, h). No cleavage of thrombin was observed with any of the treatments (Fig. 4g, i).

To determine whether *E. faecalis* proteases could functionally disrupt fibrinolysis by targeting plasmin, we incubated purified plasmin with WT and mutant strains' supernatants, followed by a secondary incubation with thrombin-generated fibrin (Fig. 4j) or Pg⁻/⁻ homogenates (Fig. 4k) that were catheterized, non-infected, for 24 h (Supplementary Fig. 7). Supernatants-containing SprE (WT and Δ*gel*E)

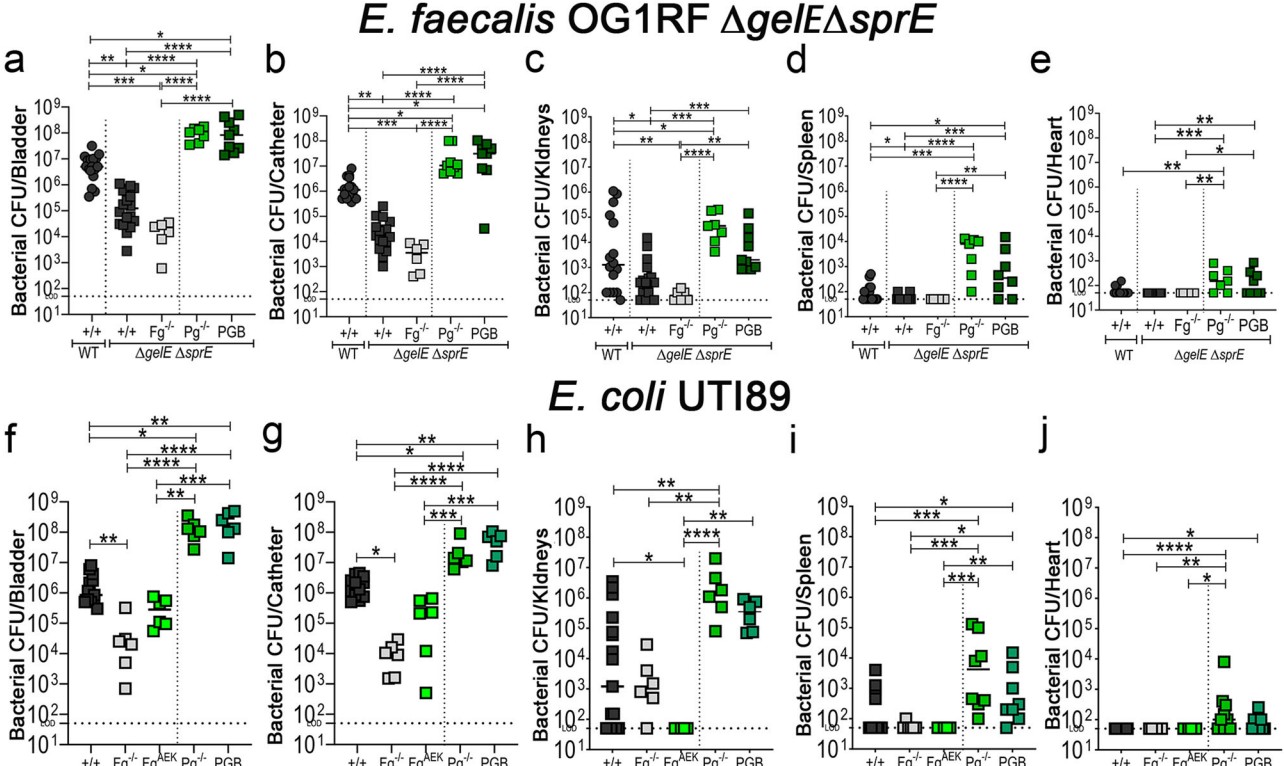

**Fig. 5 | Host fibrinolytic deficiency rescued *E. faecalis* ΔgelEΔsprE colonization deficiency, enhance colonization of uropathogen *E. coli* UTI89, promoting systemic dissemination. a–e** C57BL/6 WT and coagulation deficient mice were infected with ~2×10⁷ CFU of *E. faecalis* OG1RFΔgelEΔsprE or (**f–j**) uropathogenic *E. coli* UTI89. After 24 hpi, bacterial burdens were measured in bladder tissues (**a, f**), catheters (**b, g**), kidneys (**c, h**), spleen (**d, i**), and hearts (**e, j**). The Kruskal–Wallis test followed by a two-tailed Dunn's test was used to determine significance; *P < 0.05 was considered statistically significant. **P < 0.005; ***P < 0.0005; ****P < 0.0001. The horizontal bar represents the median value. The horizontal broken line represents the limit of detection of viable bacteria. LOD limit of detection. For CFU enumeration, infections were done at least in 3 independent experiments with n = 3–6 mice depending on the mouse genotype, and data are shown as the bacterial CFU/organ or catheter. Total n counts for **a–e**: are n = 17 for WT mice with WT *E. faecalis*; n = 18 for WT mice and double mutant *E. faecalis*; n = 6 for Fg⁻/⁻; n = 7 for Pg⁻/⁻; and n = 9 for PGB. Total n counts for **f–j**: are n = 13 for WT; n = 6 for Fg⁻/⁻; n = 6 for Fg^AEK; n = 6 for PG⁻/⁻; and n = 6 for PGB. Animals that lost the catheter were not included in this work. All n values, exact p-values, and source data are provided in Source Data file.

degraded plasmin, resulting in lack of fragment E generation, demonstrating plasmin inhibition (Fig. 4j–m). These data suggests that *E. faecalis'* SprE protease activity in urine targets the fibrinolytic system by inactivating Pg and plasmin, which results in dysregulated fibrin accumulation.

### Fibrin accumulation rescues *E. faecalis* ΔgelEΔsprE colonization deficiency

If enterococcal protease actively modulates Fg and fibrin levels to enhance CAUTI colonization, then the colonization defect observed in the absence of protease activity should be reversed in mutant mice that accumulate fibrin. Expectedly, defective colonization of *E. faecalis* ΔgelEΔsprE was further reduced in Fg⁻/⁻ mice. Predictively, defective colonization was rescued in mice that accumulate fibrin due to fibrinolytic deficiencies (Pg⁻/⁻ and PGB) (Fig. 5a, b), exhibiting higher than the WT strain colonization. Importantly, fibrin accumulation in the catheterized bladder promoted *E. faecalis* ΔgelEΔsprE systemic dissemination (Fig. 5c–e). This further demonstrates *E. faecalis'* ability to modulate the coagulation cascade, creating an environment conducive to its colonization.

### Hosts with fibrinolytic deficiencies enhance uropathogen *E. coli* UTI89 CAUTI colonization and systemic dissemination

Recently, we found that using a novel liquid-infused silicone catheter material to reduce Fg deposition on urinary catheters resulted in decreased bladder and catheter colonization and systemic dissemination by uropathogenic *E. coli* UTI89 in our mouse CAUTI

model[19]. Here, we found that bladder and catheter colonization in Fg⁻/⁻ mice or coagulation-deficient mice (Fg^AEK; soluble Fg) was significantly reduced (Fig. 5f, g). Conversely, not only was UTI89 bladder and catheter colonization significantly increased (~2 logs) in mice with fibrinolytic deficiencies (Pg⁻/⁻ and PGB) (Fig. 5f, g), but these mice also experienced significant systemic dissemination (Fig. 5h–j). These data provide evidence that uropathogenic *E. coli* colonization is modulated by Fg/fibrin in the catheterized bladder.

### Antifibrinolytic agent promotes a persistent CAUTI and systemic dissemination by diverse uropathogens

Antifibrinolytic agents including tranexamic acid (TXA) are often used in patients with bleeding disorders or patients undergoing surgical procedures[43,44] (Supplementary Table 3). TXA is extensively used to treat patients with postpartum hemorrhages, traumatic injury, and surgical procedures that increase the risk of bleeding[44,45]. TXA is a synthetic anti-fibrinolytic amino acid that acts by competitively blocking the lysine binding sites on plasmin(ogen), inhibiting plasmin interaction with fibrin[45] (Fig. 6a). As 86% of surgery patients require urinary catheterization[46], we assessed whether TXA treatment during urinary catheterization increased fibrin accumulation in the bladder, enhancing colonization and systemic dissemination by three of the most prevalent CAUTI uropathogens: *E. faecalis*, *E. coli*, and the fungal pathogen *C. albicans*[6,47]. Mice were treated with TXA or vehicle control (PBS) intraperitoneally (Fig. 6b). Then, mice were catheterized and infected with the respective pathogen and sacrificed at 1 dpi. TXA-treated mice exhibited a significant increase of bladder and catheter

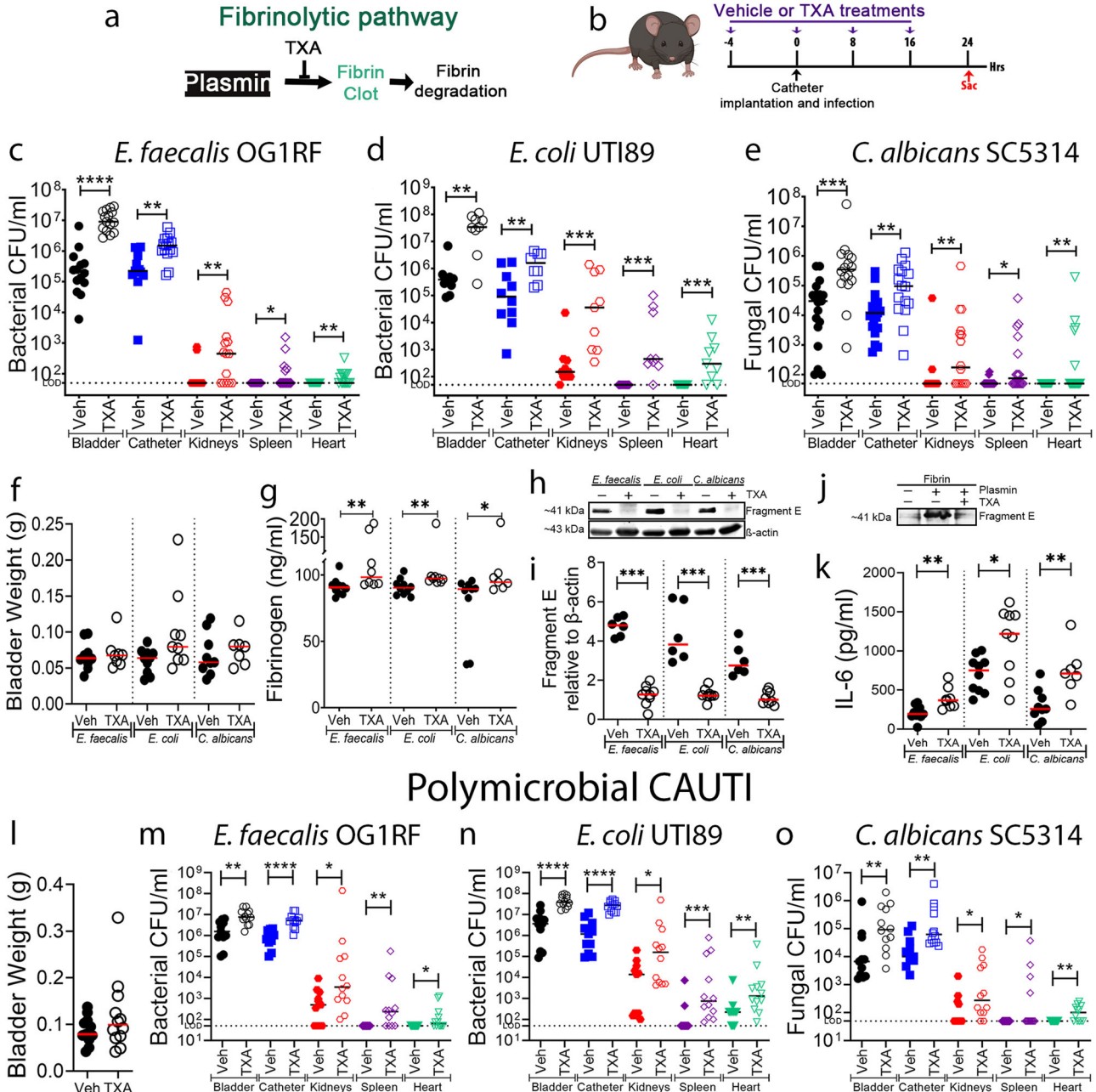

**Fig. 6 | Pharmacological inhibition of plasmin proteolytic activity inhibits fibrin degradation and further enhances pathogen burden and dissemination by three highly prevalent CAUTI pathogens, *E. faecalis*, *E. coli*, and *C. albicans*, during mono- or polymicrobial infections. a** Tranexamic acid (TXA) targets plasmin, thus inhibiting the fibrinolytic cascade. **b** Treatment timeline. C57BL/6 WT mice were infected with -2 × 10⁷ CFU of *E. faecalis* OG1RF (**c**),-2 × 10⁷ CFU of *E. coli* UTI89 (**d**), or -1 × 10⁶ CFU of *C. albicans* SC5314 (**e**) and dosed with either TXA (100 mg/mL i.p.) or vehicle (PBS). After 24 hpi, pathogen burdens were enumerated in bladder tissues, catheters, kidney pairs, spleens, and hearts. **f** Bladder edema was assessed after tissue harvest by weighing bladders. **g–i** Bladder homogenates were diluted (1:10) and analyzed for (**g**) fibrinogen, (**h**) fibrin degradation, and (**k**) IL-6. Fibrinogen and IL-6 levels were analyzed via ELISA. Fibrin degradation in the bladder homogenates was analyzed by measuring fragment E production via western blot and β-actin was used as loading and normalization control (**h**) and quantified by densitometry (**i**). TXA inhibition of plasmin-dependent fibrin degradation was further confirmed and analyzed in vitro using purified human proteins via western blot (**j**). **l–o** Polymicrobial CAUTI with *E. faecalis*, *E. coli*, and *C. albicans*. **l** Bladder edema of mice treated with TXA or vehicle during polymicrobial CAUTI.

**m–o** C57BL/6 WT mice were infected with -2 × 10⁷ CFU of *E. faecalis* OG1RF (**m**),-2×10⁷ CFU of *E. coli* UTI89 (**n**), and -1 × 10⁶ CFU of *C. albicans* SC5314 (**o**) and dosed with either TXA (100 mg/mL i.p.) or vehicle (PBS). Pathogen burden was assessed after 24 hpi in bladder tissues, catheters, kidney pairs, spleens, and hearts. The Mann–Whitney U test was used to determine significance; *$P < 0.05$ was considered statistically significant. **$P < 0.005$; ***$P < 0.0005$; ****$P < 0.0001$. The horizontal bar represents the median value. The horizontal broken line represents the limit of detection of viable pathogen. LOD limit of detection. For CFU enumeration, infections were done at least in 3 independent experiments with $n = 3–6$ mice each one, and data are shown as the microbial CFU/organ or catheter. For **c**: $n = 14$ for vehicle and $n = 15$ for TXA. For **d**: $n = 10$ for vehicle and $n = 9$ for TXA. For **e**: $n = 19$ for vehicle and $n = 16$ for TXA-treated. For **f**: $n = 10$ for vehicle + *E. faecalis*, $n = 8$ for TXA + *E. faecalis*, $n = 10$ for vehicle + *E. coli*, $n = 9$ for TXA + *E. coli*, $n = 8$ for vehicle + *C. albicans*, and $n = 7$ for TXA + *C. albicans*. For **i–k**: $n = 6$ for all vehicle and $n = 9$ for all TXA. For **l–o**: $n = 11$ for all vehicle and $n = 12$ for all TXA-treated. Animals that lost the catheter were not included in this work. All $n$ values, exact p-values, and source data are provided in Source Data file.

colonization and promoted systemic dissemination of all pathogens when compared with PBS-treated mice (Fig. 6c–e). No significant differences in bladder weights between treatments were observed, suggesting that TXA did not exacerbate catheter-induced inflammation (Fig. 6f). To confirm fibrinolysis inhibition and fibrin accumulation, Fg/fibrin concentration and fragment E generation was compared and quantified between PBS- and TXA-treated bladders (Fig. 6g–i) and further confirmed in vitro (Fig. 6j). We found that Fg/fibrin concentration increased and fragment E generation decreased in TXA-treated bladders (Fig. 6g–i). Since Fg induces IL-6 expression in immune and epithelial cells[48,49], we measured IL-6 levels in the bladder. We found that IL-6 levels significantly increased in TXA-treated mice (Fig. 6k). This is consistent with increased Fg/fibrin levels in the TXA-treated bladders, suggesting a positive-feedback loop between the inflamed bladder and Fg in the catheterized host.

We further tested whether fibrin accumulation could enhance polymicrobial CAUTI. For this, TXA and PBS-treated mice were catheterized and infected with all three uropathogens, *E. faecalis*, *E. coli*, and *C. albicans*[6,47]. We found no difference in inflammation between PBS- and TXA-treated mice (Fig. 6l). However, blocking fibrinolysis with TXA significantly enhanced the burden of all pathogens in the bladders and catheters and increased systemic dissemination (Fig. 6m–o). In conclusion, these data show that pharmacological inhibition of fibrinolysis in the host enhances mono- and polymicrobial CAUTI and systemic dissemination by diverse uropathogens.

## Discussion

CAUTIs remain prevalent and costly, increasing patient morbidity and mortality[50,51]. CAUTI prevention and treatment faces major challenges, including the ability to predict: i) when catheterized patients develop infections; ii) which uropathogen will cause the infection; and iii) which patient population is more susceptible to septicemia. Previous attempts at treating CAUTIs include vaccine development, antibody therapy, and compounds that target specific pathogens[6,52]. However, multiple pathogens from different kingdoms can simultaneously cause CAUTIs and the risk of developing an infection increases with dwelling time. Efficiently predicting when patients will develop an infection, the causative organism, and which patient populations are more susceptible to systemic dissemination is imperative for successful CAUTIs prevention and treatment. Thus, a better understanding of CAUTI pathophysiology is critical to develop better clinical practices for improving patient outcomes.

Here, we found that the coagulation cascade plays an important role in CAUTI outcome. Importantly, we identified that host coagulopathies that result in fibrin clot accumulation promote higher pathogen colonization in the catheterized bladder by the most prevalent uropathogens in mono- or poly-microbial infections, enhancing persistence during CAUTI and promoting bloodstream infection and systemic dissemination (Figs. 3, 5, and 6). Fibrin is a major thrombus-formation end product[53] and its concentration is tightly controlled by a series of cofactors, inhibitors, and receptors[15]. Once healing is resolved, the fibrinolytic pathway is triggered by activation of Pg into plasmin by uPA and tPA, promoting fibrin clot degradation and restoring tissue homeostasis (Fig. 3a)[14,15]. In the catheterized-bladder, our results show that uPA, which was originally isolated from human urine, is the main Pg activator, but not tPA (Fig. 3). This could be related to their action mechanisms as tPA activation of Pg requires attachment to the fibrin on the clot surface to activate fibrin-bound Pg[54]. Since CAUTI pathogens form biofilms on fibrin, they may occlude tPA binding. In contrast, uPA is fibrin-independent and activates Pg in solution or when associated with its cellular receptor uPAR[55], providing a cell-mediated Pg activation[56]. uPAR is present on many immune cells including macrophages[56] that are recruited by urinary catheterization[20,27,28]; thus, it is possible that these immune cells are important for uPA-cell-mediated Pg

activation. The uPA activation mechanism of Pg will require further experimentation.

Hypofibrinolysis and several thromboembolic diseases, including strokes, venous thromboembolism, rheumatoid arthritis, and renal diseases are accompanied by fibrin accumulation[57,58]. Furthermore, there are several congenital deficiencies that result in fibrin clot accumulation, including individuals with deficiencies of Pg, tPA, uPA, PAI-1, PAI-2, TAFI, and Annexin A2[53] (Supplementary Table 3). Consistent with our CAUTI data, fibrin accumulation from imbalance between activation of coagulation and inhibition of fibrinolysis, is detrimental for patients suffering from COVID-19[59] and ventilator-associated pneumonia[60]. Moreover, fibrin accumulation can also result from antifibrinolytic treatments, which are commonly use in patients with bleeding disorders[58,61] (Supplementary Table 3), in patients with traumatic injuries, or undergoing surgical procedures that increase bleeding risk, including liver transplants, cardiac and cesarean surgeries[43–45,62]. TXA is an active inhibitor of uPA and tPA, and also plasmin-fibrin binding[63,64]. Importantly, TXA elimination from the body is by urinary excretion, increasing its concentration in urine[62,63]. This information further validates our results showing that TXA is an efficient antifibrinolytic agent in the catheterized bladder, resulting in fibrin accumulation and enhancing CAUTI and higher systemic dissemination. Based on our data, these patient populations with coagulopathies or treatments that result in fibrin accumulation, if receiving a urinary catheter, may have a higher risk of developing a CAUTI and could be more susceptible to secondary bacterial and fungal infection from a CAUTI.

During catheterization, IL-6 is the most highly elevated cytokine in the bladder, bloodstream, and liver and *E. faecalis* infection promotes its concentration (Fig. 1i). IL-6 is a potent regulator of inflammation, coagulation, and the immune response[65] by regulating neutrophil infiltration, monocytes to macrophages differentiation, and Fg expression by hepatocytes[66,67]. Interestingly, Fg/fibrin also stimulates IL-6 expression by immune cells in the damaged tissue[56,68]; thus, resulting in a positive-feedback loop, dysregulated inflammation, and Fg/fibrin accumulation, promoting microbial colonization and perpetuating a futile inflammation cycle that benefits the pathogen. Importantly, IL-6/Fg dysregulation play a role in chronic inflammatory diseases and cancer[30,31,66]. IL-6 pathway activation was further supported by Metascape analysis showing that CEBPA and STAT3 are highly predicted transcriptional regulators of the catheterized bladder environment (Supplementary Fig. 5b). IL-6 levels has been associated with severe and febrile uUTI[69]. However, during early, recurrent, or asymptomatic uUTI, IL-6 is repressed, unchanged or low levels[70–74]. In fact, TNFα is elevated during acute uUTI and is associated with recurrent uUTI[73,74]. Importantly, TNFα is repressed during CAUTI (Fig. 1h, Supplementary Fig. 3). This further demonstrates that uUTI and CAUTI have different pathophysiologies. Therefore, targeting IL-6/Fg signaling to modulate its effects on pathogenesis may provide a novel approach to improving CAUTI outcomes.

Here, we found that SprE actively degrades Pg and plasmin, suggesting that *E. faecalis* uses SprE to inactivate the fibrinolytic system. This strategy differs from other bacterial pathogens, including Group A Streptococcus (GAS), which activates Pg into plasmin via streptokinase, dissolving blood clot to cause an invasive infection[75]. Importantly, SprE specificity against Pg, Pm, and prekallikrein, suggests that SprE is recognizing a specific recognition site. Therefore, identifying Pg/plasmin or prekallikrein cleavage sites using purified SprE will be important to determine its action mechanism. Furthermore, it would be interesting to understand if other uropathogens produce secreted proteases with similar SprE activity to target the fibrinolytic system to enhance their colonization.

*E. faecalis* induction of fibrin accumulation creates ideal conditions for biofilm formation and persistence in the bladder, which is an

open and dynamic system, where urine is constantly passing. Fibrin accumulation also contribute to the Fg/IL-6 feedback loop. This further explains how *E. faecalis* can thrive and exploit the catheterized bladder despite robust inflammatory responses and how fibrin accumulation is also conducive for CAUTI and systemic dissemination by other pathogens. Enterococci's ability to modulate the bladder environment to benefit itself and other pathogens may further explain why enterococci are described as a pioneer species on urinary catheter[23]. This supports several clinical studies that showed enterococci are early colonizers and persist during prolonged catheterization in mono-microbial or polymicrobial interactions[5,23,76].

In uUTI, *E. coli* uses the type 1 pilus to bind mannosylated-receptors on the urothelial surface to invade and form intracellular bacterial communities (IBCs)[52]. During *E. coli* CAUTI, IBC formation is significantly reduced without affecting overall bladder colonization[77,78]. We found that Fg, which is highly glycosylated, is used by *E. coli* colonize the bladder and catheter during CAUTI[19]. Fg deposition onto the urothelium could block *E. coli*-cell interaction, affecting IBC formation. Consistent with this, we found that Fg/fibrin levels directly affected *E. coli* CAUTI (Fig. 5f, g), decreasing colonization when Fg was soluble or absent while increasing colonization when fibrin accumulated. Soluble Fg and other host glycosylated proteins, including uromodulin[78], may act as decoy receptors to prevent binding to urinary tract surfaces, resulting in the pathogen's expulsion by urine flow. In contrast, fibrin accumulation on surfaces can provide a platform for *E. coli* adherence to counteract its clearance by urine flow.

Given how common urinary catheterization is and the rise of multidrug resistant pathogens, the CAUTI frequency is expected to keep increasing. This study not only has identified mechanisms by which catheter-induced inflammation predispose patients to development of CAUTIs but also identified host populations that will be at higher risk of a CAUTI-associated septicemia caused by the most prevalent CAUTI pathogens (Supplementary Fig. 9). In our comparative catheter proteomic analysis between human and mouse, we found our human catheters had less abundance of deposited proteins than in mouse catheters. This could be explained by stochastic nature of protein deposition due to urine flow or by catheter removal from the host. For example, patient catheter removal requires pulling the catheter through the urethra, which may randomly strip some proteins off. Despite this, the shared proteins are involved in the same host pathways, further confirming that our mouse CAUTI model has shown to faithfully recapitulate the pathophysiology of human CAUTI, suggesting that our findings have the potential to be translated for prevention and management of human CAUTI[6,17,18,21,52]. Leveraging these results to develop improved strategies for at-risk patient identification and to inform catheterization guidelines, will provide a higher patient quality of life and minimize the risk for complications.

## Methods

### Ethics statement
All animal care was consistent with the Guide for the Care and Use of Laboratory Animals from the National Research Council. The University of Notre Dame Institutional Animal Care and Use Committee approved all mouse infections and procedures as part of protocol #22-01-6971. For urine collections, all donors signed an informed consent form and protocol was approved by the Institutional Review Board of the University of Notre Dame under study #19-04-5273.

### Mouse handling and husbandry
Mice used in this study were ~6-week-old female wild-type C57BL/6 mice purchased from Jackson Laboratory and National Institute of Cancer Research or mutant mice bred in Harper Cancer Research Institute (Notre Dame, Indiana). The University of Notre Dame Institutional Animal Care and Use Committee approved all mouse

infections and procedures as part of protocol number 22-01-6971. All animal care was consistent with the Guide for the Care and Use of Laboratory Animals from the National Research Council.

### Mouse infection models
Mice were catheterized by placing a silicone catheter in the bladder through the urethra (transurethral) and infected as previously described[79]. Briefly, mice were anesthetized with isoflurane (inhalation) and implanted transurethrally with a 6-mm-long silicone tubing which engulfed a 4mm-long PE tubing (Table 1). Mice were transurethrally infected with 50 µl of respective pathogen at an $OD_{600}$ of 0.6 (~$2 \times 10^7$ CFU) or fungi at an $OD_{600}$ of 5.0 (~$1 \times 10^6$ CFU) in PBS, or only catheterized. In triple infection experiments, the final CFUs/ 50 µl were ~$2 \times 10^7$ CFU for bacteria and ~$1 \times 10^6$ CFU for fungi. Immediately before sacrifice, blood was harvested via cheek poke and serum was extracted from whole blood (BD Microtainer, Table 1) for cytokine analysis as described below. To harvest catheters and organs, mice were sacrificed at their specified time-point by cervical dislocation after anesthesia inhalation; the bladder, kidneys, heart, spleen, liver, and if present, silicone catheter were aseptically harvested and bladders weighed. Except livers, all organs were homogenized (1 min shake, 5 min rest, 1 min shake; MP Biomedical 116005500) and plated for CFU enumeration as described below. A subset of bladders used for histology analysis were weighed, fixed and processed as described below. Homogenized samples of bladder, liver and blood serum were sent for cytokine analysis as described below. Catheters were subjected to 15 min sonication (Branson 2800, Table 1) for CFU enumeration or sent for proteomic analysis as described below using nonimplanted catheters as controls. The University of Notre Dame Institutional Animal Care and Use Committee approved all mouse infections and procedures as part of protocol number 22-01-6971. All animal care was consistent with the Guide for the Care and Use of Laboratory Animals from the National Research Council[79].

### Microbial strains and growth conditions
Microbe strains used in this paper include *E. faecalis strains*, uropathogenic *E. coli*, and fungal *C. albicans* details listed in Table 1. Unless otherwise noted, all *E. faecalis* strains were grown overnight in 10 mL of brain heart infusion (BHI) media (Hardy Diagnostics C5143) supplemented with 50 µg/ml of rifampicin and 50 µg/ml of fusidic acid at 37 °C in static conditions. *C. albicans* was grown in overnight in 10 mL of yeast-peptone-dextrose (YPD) media supplemented with 500 µg/ml of kanamycin at 37 °C in static conditions. *E. coli* UTI89 strain was grown in Luria-Bertani (LB) medium (MP Biomedicals) supplemented in 100 µg/ml of kanamycin under shaking at 37 °C for 4 h. Then, diluted in LB+kan (1:1000) then grown for 24 h in static conditions, then diluted once more in LB+kan (1:1000) and grown again for 24 h in static conditions. For differential selection and growth of the pathogens, *Candida albicans* was grown in YPD agar plates containing 500 µg/ml of kanamycin, a concentration that will exclude both *E. faecalis* and *E. coli* growth. *C. albicans* grows very well in YPD media due its acidic pH, and *C. albicans* growth is restricted on LB and BHI media. *E. faecalis* was selected on BHI plates supplemented with 50 µg/ml of rifampicin and fusidic acid; these antibiotic concentrations inhibit *E. coli* growth. *E. faecalis* growth is restrictive in YPD and LB media. Verification of the strains was further confirmed by their distinctive colony morphologies.

### Human urine collection
Human urine was collected and pooled from a minimum of two healthy female donors between the ages of 20–35 years to minimize donor variability. Donor health was indicated through lack of kidney disease, diabetes mellitus, or recent antibiotic treatment. Urine was filter-sterilized with a 0.22 µm filter (VWR 29186-212) and normalized to a pH

**Table 1 | Key resources**

| Reagents or resource | Source | Identifier |
|---|---|---|
| *Bacterial and fungal Strains* | | |
| *Enterococcus faecalis* OG1RF | Murray et al.[84] | N/A |
| *Enterococcus faecalis* ΔsprE | Xu et al.[29] | N/A |
| *Enterococcus faecalis* ΔgelE | Xu et al.[29] | N/A |
| *Enterococcus faecalis* ΔsprE/ΔgelE | Xu et al.[29] | N/A |
| *Escherichia coli* UTI89 | Mulvey et al.[85] | N/A |
| *Candida albicans* SC5314 | Odds et al.[86] | N/A |
| *Chemicals, peptides, and other* | | |
| Anti-beta Actin antibody | Abcam | ab8229 |
| Anti-fibrinogen antibody | Abcam | ab34269 |
| Donkey Anti-Goat IgG Polyclonal Antibody | LI-COR Biosciences | 925-68074 |
| Donkey Anti-Rabbit IgG Polyclonal Antibody | LI-COR Biosciences | 926-32213 |
| Human Endoplasmin | VWR | 102830-698 |
| Human Factor XII | Enzyme Research Laboratories | HFXII 1212 |
| Human Fibrinogen | Enzyme Research Laboratories | Fib 3 |
| Human Fibrinogen ELISA Kit | Abcam | ab241383 |
| Human Plasmin | Enzyme Research Laboratories | HPlasmin |
| Human Plasminogen | Enzyme Research Laboratories | HPg 2001 |
| Human Prekallikrein | Enzyme Research Laboratories | HPK 1302 |
| Human Thrombin | Sigma-Aldrich | T6884-250UN |
| Plasminogen polyclonal antibody | Proteintech | 17462-1-AP |
| Thrombin polyclonal antibody | Fisher-Scientific | PA5-99213 |
| Tranexamic Acid | VWR | TCA0236-100G |
| IL-6 Mouse Uncoated ELISA Kit | Invitrogen | 88-7064-22 |
| Bio-Plex 23-Plex Assay Kit, Mouse | Bio-Rad | M500KCAF0Y |
| Bio-Plex 27-Plex Assay Kit, Human | Bio-Rad | M60009RDPD |
| *Organisms: mutant mice* | | |
| C57BL/6 mice | NCI/Jax | 000664 |
| Fg$^{-/-}$ | Ploplis et al.[35] | N/A |
| Fg$_{AEK}$ | Prasad et al.[36] | N/A |
| Pg$^{-/-}$ | Ploplis et al.[35] | N/A |
| uPA$^{-/-}$ | Carmeliet et al.[40] | N/A |
| tPA$^{-/-}$ | Carmeliet et al.[40] | N/A |
| FVII$_{tTA}$ | Rosen et al.[37] | N/A |
| PAI$^{-/-}$ | Carmeliet et al.[41] I/II | N/A |
| PGB | Iwaki et al.[42] | N/A |
| *Equipment* | | |
| Zeiss Axio Observer | Zeiss | https://www.zeiss.com/microscopy/en/products/ |
| FastPrep-24™ 5G | MP Biomedical | 116005500 |
| Odyssey CLx Imager | LI-COR Biosciences | https://www.licor.com/bio/support/answer-portal/imaging-systems/odyssey-clx.html |
| Sonicator | Branson | 2800 |
| Spectramax ABS plus | Molecular Devices | ABSPLUS |

of 6.0–6.5. All participants signed an informed consent form and protocols were approved by the local Internal Review Board at the University of Notre Dame under study #19-04-5273.

### Collection of human urinary catheters
Patient catheters were collected with informed consent after the clinical decision to remove for standard of care was made as described[80]. This study was approved by the Washington University School of Medicine (WUSM) Internal Review Board (approval #201410058) and performed in accordance with WUSM's ethical standards and the 1964 Helsinki declaration and its later amendments. Moreover, 10 catheters that had a dwell time of 24–26 h were further processed by the S.J.H. and M.G.C. laboratories for proteomics analysis.

### H&E staining of mouse bladders and imaging
Mouse bladders were fixed in 10% neutralized formalin for 18 h, embedded, sectioned, and stained as previously described[21]. Briefly, bladder sections were deparaffinized, rehydrated, and rinsed with water. Hematoxylin and Eosin (H&E) stain for light microscopy was done by the CORE facilities at the University of Notre Dame (ND CORE). All imaging was done using a Zeiss Axio Observer inverted light microscope. Zen Pro and ImageJ software were used to analyze the images.

## Microbial enumeration

Microbe load from bladder, kidney, spleen, hearts and silicone catheters from sacrificed animals was determined via serial dilution and enumeration. Organs were homogenized and silicone catheters were cut in small pieces before sonicated for CFU enumeration. Pathogens were plated in their corresponding media conditions (see *Microbial strains and growth conditions*).

## Cytokine analysis

Bladder, liver, and blood serum samples from mice catheterized and infected with *E. faecalis* OG1RF or catheterized and mock infected with PBS for 1 h, 3 h, 6 h, 9 h, 12 h, 1d, 3d, 7d, and 14d were frozen at −80 °C until time of assay. Before cytokine analysis, homogenates were thawed on ice and microcentrifuge at 11,000 × *g* for 10 min, and supernatants transferred to a new tube. Human urine was collected from patients catheterized for about 24 h. Both mice and human samples were probed for IL-6, IL-1α, IL-1β, and TNFα levels using a Bio-Plex Multi-Plex Assay Kit from Bio-Rad Laboratories (Table 1) following the manufacturer's protocols.

## Proteomic analysis of catheters

Donated catheters from patients catheterized for 24–26 h, or harvested catheters from mice that were catheterized and infected for 1 h, 3 h, 6 h, 1 day, 3 days, 7 days, and 14 days were boiled in SDS-sample buffer to release proteins. Protein samples were then processed by Proteomics and Mass Spectrometry Facility at the Danforth Plant Science Center (St. Louis, MO) as previously described[57]. Briefly, protein samples were trypsin-digested overnight at 37 °C. The extracted peptides from each sample were dried and resuspended in 100 μL 1% ACN/1% FA. 5 μL were processed with an LTQ-Orbitrap Velos Pro on a Dionex RSLCnano HPLC using a 1 h gradient. All MS/MS samples were analyzed using Mascot (Matrix Science, London, UK; version 2.5.1.0). Mascot parameters were set to search the cRAP_20110301 and NCBInr databases (selected for *Mus musculus* or *Homo sapiens* depending on catheter host) with a trypsin-digestion assumption. MS/MS peptide and protein identifications were validated with Scaffold (version Scaffold_4.4.5 Proteome Software Inc., Portland, OR). Protein identifications of mouse and human catheters that were for 1 dpi (mice) and 24–26 h dwell time (humans) were compared using an E Venn network[81] online software (http://www.ehbio.com/test/venn/#/) to identify common proteins in the catheterized bladder environment between mice and humans. The 76-shared proteins were then submitted to metascape.org (Supplementary Table 1) for gene ontology classification, gene ontology enrichment, protein-protein interaction identification, network analysis, and prediction of transcriptional regulatory relationships[34]. Generated networks were modified using Cytoscape software[82].

## Bacterial protease activity assay

Cultures of *E. faecalis* strains were centrifuged for 10 min at 3000× and washed three times with 1× PBS. Pellets were resuspended in 10 ml of filter sterilized human urine at pH 6.5. The resuspended cultures were statically incubated for 24 h at 37 °C. Following incubation, cultures were centrifuged and the supernatants were collected, filter-sterilized, and concentrated and size excluded using Macrosep Advance Centrifugal Devices (10k molecular weight cut-off; PALL MAP010C37). 40 μg/mL of purified protein (hPlasmin, hPlasminogen, hThrombin, hEndoplasmin (HSP90b1), hPrekallikrein, or hFactor XII) were added to supernatants and incubated statically for 4 h at 37 °C. To determine proteolytic activity of *E. faecalis* strains in mice, urine-cultured supernatants were also added to PGB mouse bladder homogenates (1:5) that were catheterized only for 24 h. The mixture was incubated for 4 h at 37 °C. Following incubation, 5× SDS sample buffer was diluted to 1× into the samples and boiled at 95 °C for 2 min before separation on 12% acrylamide gels at 100 V. Plasmin that was incubated with *E. faecalis*

mutant strains' supernatants was subsequently incubated with 25 μl of thrombin-generated fibrin (from 2 mg/mL of purified fibrinogen) or 25 μl of 24-h-post catheterization (hpc) Pg$^{-/-}$ bladder homogenates for 4 h at 37 °C to determine functional disruption of fibrinolytic activity. Fibrin and homogenates incubated with supernatant-treated plasmin was then also processed for acrylamide gel protein separation. Graphical methods provided in Supplementary Fig. 7. Normalization and western blots were done as described below.

## Western blot

Western blotting was done as previously described[83]. Briefly, bladder homogenate samples were diluted 1:1 into 5X sample buffer then boiled at 95 °C for 2 min. Following boiling, 10 uL of sample in SDS was run on a 12% polyacrylamide gel for separation. Gels were then stained with Coomassie Blue for 60 min followed by de-staining and imaging on an Odyssey infrared imaging system (LI-COR Biosciences, Lincoln, NE) to normalize protein concentrations. Following normalization, samples were again run on 12% acrylamide gels at 4 °C and 50 V. The samples were then transferred to polyvinylidene difluoride membrane using a semi-dry transfer (Millipore Sigma Cat# IPFL00005, Table 1). Following transfer, membranes were blocked in 5% skim milk and probed by Western immunoblot with anti-β-actin (Abcam ab8229; 1:10,000) as a loading control, and anti-plasminogen primary antibodies (Proteintech 17462-1-AP; 1:1000, respectively), anti-thrombin (ThermoFisher Scientific PA5-99213; 1:1000) or anti-Fg (Abcam ab34269; 1:1000) diluted in 3% skim milk in Tris-buffered saline-Tween (.2%). Membranes were then probed with secondary antibodies Donkey anti-goat (LI-COR Biosciences 925-68074) and Donkey anti-rabbit (LI-COR Biosciences 926-32213), diluted in dilution buffer (3% skim milk in .2% TBS-T, and .01% SDS) and visualized on an Odyssey CxL.

## Measurement of Fg and IL-6 from bladder homogenates

Bladder homogenates from respective mouse bladder groups were analyzed to determine abundances of Fg and IL-6 cytokine with respective ELISA kits following manufacture-provided protocols. Briefly, bladder homogenates were centrifuged and supernatants were diluted 1:10 in ELISA diluent. Dilution standard preparation was performed following manufacturer guidelines. Standards/samples were then placed on to pre-coated ELISA plates overnight at 4 °C, removed, plates washed three times with wash buffer, incubated with detection antibodies for 1 h at room temperatures, washed three times, incubated with appropriate biotinylated-detecting enzyme for 30 min at room temperature, washed 5 times, and incubated with appropriate chromogenic substrate for 15 min at room temperature, and stopped with appropriate stop solution. Absorbances were measured at appropriate wavelengths with a plate reader (Molecular Devices Spectramax ABS plus).

## Measurement of fragment E

Fragment E was probed via SDS-PAGE and western blot with an anti-Fg (1:1000) primary antibody and anti-β-actin (1:10,000) as a loading control. Fragment E probing was performed for the following experiments: i) in vitro bacterial protease activity of plasmin, previously incubated with urine-grown *E. faecalis* supernatant, incubated with fibrin or with Pg$^{-/-}$ homogenates (Fig. 4j–m), ii) homogenized bladders from both TXA-treated and vehicle controls (Fig. 6h), and iii) in vitro inhibition of plasmin fibrinolytic activity by TXA (Fig. 6i). Fragment E band was visible at ~41 kDa. Measurements of the median fluorescent intensity from each sample of both β-actin and fragment E lane were taken using image studio to determine densitometry. Measurements were graphed into Graphpad Prism 9 for statistical analysis.

## Statistical analysis and reproducibility

Data derived from this study was entered into Graphpad Prism 9 to generate statistical results and graphs. Kolmogorov–Smirnov

analytical tests and quantile-quantile (QQ) plot visual normality tests were performed to assess data distributions which, based on results, determined the appropriate parametric or non-parametric test to use. Normally distributed data was tested for difference between two groups with a student T-test, while an ANOVA followed by a Tukey's post-hoc was used to determine statistical differences between more than two groups. For non-normal distributions, a Mann–Whitney U test was used to determine statistical difference between two groups, while a Kruskal–Wallis test followed by a Dunn's test was used to determine significance between more than two groups. Pearson's correlation statistical analysis was used to measure association between variables. Statistical tests used are indicated in figure legends.

### Reporting summary

Further information on research design is available in the Nature Portfolio Reporting Summary linked to this article.

## Data availability

Proteomics raw data are available in Supplementary Data 1 and 2 files. The mass spectrometry proteomics data have been deposited to the ProteomeXchange Consortium via the PRIDE partner repository with the dataset identifier PXD050199 and 10.6019/PXD050199. The data generated in this study are provided in the Source Data file. Source data are provided with this paper.

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

## Acknowledgements

We thank the Freimann Life Science Center for mouse breeding and husbandry. Graphical illustrations were created with biorender.com. This work was supported by institutional funds from the University of Notre Dame (to A.L.F.M.), from Good Venture Foundation (Open Philanthropy) grant (to A.L.F.M., J.J.M, K.K., and M.J.A.), National Institutes of Health grants R01DK128805 (to A.L.F.M., J.J.M, K.K., and M.J.A.), R21AI163825 and U19AI157797 (to M.G.C and W.X.); R01-HL013423 (to D.D., V.A.P., and F.J.C.); R01-HL160046 and U01-HL143403 (to M.J.F); and DK051406 (to A.K., K.B., A.D, and U.D., and S.J.H). Additionally, J.J.M. has received support from the College of Science's Paul F. Ware, M.D. Graduate Fellowship.

## Author contributions

A.L.F.M. conceived and supervised the research. J.J.M., K.K., M.J.A., C.G., E.W., E.R.L., W.X., A.P., and A.L.F.M. conducted the experiments and sample and data analyses. A.L.F.M., A.K., K.B., A.D. conducted human sample collection. A.L.F.M., J.J.M., K.K., and M.J.A. wrote the manuscript. A.L.F.M., J.J.M., K.K., M.J.A., A.D., W.X., D.D., M.G.C., F.J.C., V.P., M.J.F., and S.J.H. reviewed and edited the final manuscript.

## Competing interests

The authors declare no competing interests.
