## [Peer Review File · Nature Communications]

REVIEWER COMMENTS

Reviewer #1 (Remarks to the Author):

UTI is one of the most common infections worldwide, and catheter-associated UTI (CAUTI) is more complicated and often lethal due to blood stream infection (e.g., sepsis) and multi-drug resistance. The manuscript by Molina et al. reported a mechanistic study of CAUTI. The authors established several mouse models with varied genetic deficiencies (e.g., Fg^{-/-}, Pg^{-/-}, tPa^{-/-}) that may enhance or decrease CAUTI; they also developed bacterial cell lines with different mutants (e.g., *E. faecalis* Δ gelE Δ sprE) and thus varied virulence; meanwhile, they performed a number of in vivo and histological assays to examine the bacterial growth, biofilm formation, and infection persistence under CAUTI condition. In addition, the authors also reported an interesting connection of tranexamic acid (TXA) with CAUTI, and suggested an unexpected side effect of this common bleeding-control drug. The study is comprehensive and makes biological science in terms of experimental design, controls, and data interpretation. The reviewer is going to only comment on the contents that he has relevant expertise.

1. One of the major concerns is about the novelty of some of the findings. The authors should highlight the findings if they are indeed new, or justify their findings if they are known already.

For instance, fibrinogen (Fg) has been known to be a major component of urinary catheter proteome, based on the previous studies from others (PMID: 34249966) and this lab (PMID: 35348114). It is not clear the paragraph (lines 145-155) presented here is just a fact or a new finding of this study.

2. This lack of clarity is also true to a few other result paragraphs. *E. faecalis* is known to be one of the most prevalent uropathogens in catheter, and its association with Fg is known as well. The authors should describe clearly whether their data are plain facts or new findings.

3. Lines 152-153 and Figure 2b: The figure caption is not clear to the readers. What the numbers are and how they were calculated.

Reviewer #2 (Remarks to the Author):

This manuscript by Molina et al. builds on prior work showing the important contribution of fibrinogen deposition in the urinary tract to catheter-associated UTI. In the current study, one agent of CAUTI, *Enterococcus faecalis*, is shown to use a protease, SprE, to degrade plasmin/plasminogen. Experimental UTI in a variety of mice with mutations in clotting and fibrinolytic systems further demonstrates that fibrin/fibrinogen increases bacterial burden, and this was also true for another bacterial pathogen, *Escherichia coli*, and a fungal pathogen, *Candida albicans*. Finally, the authors report that an antifibrinolytic medication prescribed during surgeries, tranexamic acid, caused a higher bacterial burden during experimental UTI using the murine CAUTI model.

The findings of 1) a bacterial mechanism for manipulating host plasminogen, 2) multifactorial use of mutant mice to assess the contributions of Pg/Fg toward increasing bacterial burden during UTI, and 3) a potential major contributor to CAUTI vulnerability (routine use of anticlotting medication TXA in surgical patients) make this an exciting manuscript. Overall, this was a very interesting study that nevertheless has some major concerns that need to be addressed before publication.

Major items:

1. I have several concerns and comments on the comparisons between mouse and human data.
 - A. line 160: It's interesting that approximately twice as many proteins were identified on mouse catheters as human catheters. Naively, I would expect the much larger surface area available on patient catheters would allow more proteins to be detected. An alternative explanation is that the unanchored catheter fragment in the tiny mouse bladder is much more disruptive than a standard Foley with an inflated balloon (compare the architecture of the naïve bladder in Fig 1g to the catheterized samples). This intriguing and important proteomics comparison will help inform the strengths and limitations of this model. Do the data support this interpretation as an important caveat to understanding the catheter implantation model? These points should also be considered in the conclusion on Line 407.
 - B. line 164: Likewise, for translation to the medical setting, the human proteins that did not overlap with mice are also important. Please comment on interesting features of these proteins: are these proteins only produced in humans? Do the differences reflect more severe inflammation in the mouse catheter implantation model vs clinical catheterization?
 - C. Line 75: Substitute "mice" for "host populations." Likewise, soften or qualify line 308 "perpetual inflammatory positive-feedback loop." Human susceptibility due to genetic or pharmacologic intervention is reasonably speculated in the Discussion based on the mouse data, but no direct patient data are provided in this manuscript for the broad statement provided here. IL-6 measurement in human urine is interesting but should not be overinterpreted to make a direct connection to *E. faecalis* or fibrinogen without further investigation; for example, degradation of plasminogen in patient urine when *E. faecalis* but not other uropathogens are present. Lines 354, 362: Is there enhanced risk of CAUTI or other UTIs in people with fibrin deficiencies or other bleeding disorders?
2. Statistics. Multiple figure legends state "values represent means \pm SEM" when medians are shown without error bars (e.g., Fig. 4, line 933, or worse, lines 841-2, which confusingly state "Values represent means \pm SEM. Values represent median."). The use of medians is typical for these nonparametric datasets, but the statistics need to be checked and clarified to address the nonsensical statements about SEM. Second, the same legends that mention SEM also mention a second test, Mann-Whitney U. This makes more sense for the data presented. However, many figs (e.g., Fig 3g-h) show multiple comparisons, and Mann-Whitney U is a pairwise test. Another test such as Kruskal-Wallis with multiple comparison corrections would be more appropriate for these scenarios. Extended Data Fig. 3 shows what appears to be means \pm SEM (a measure of normal distribution) but again states that values represent medians and the non-parametric Mann-Whitney U test was used to measure significance. What test was actually used?
3. Specificity of SprE toward plasmin and plasminogen. Strong evidence is provided that SprE degrades these proteins, but the specificity toward these two proteins is overstated. First, thrombin is provided as a comparator, but it is much smaller than the other two proteins. By itself, this is not strong evidence for specificity. Inclusion of additional clotting/fibrinolytic factors, especially of comparable size, would be more indicative of specificity. Even testing non-coagulation cascade proteins of comparable size would be useful. Second, the cropped images in Fig 4 g,j,k are difficult to assess; whole blots/gels should be provided in supplemental data. The mysterious cross-reactive material in Fig 4g is not sufficient to convey this information. Fig. 4j lacks a loading control.
4. There seem to be two different messages that aren't quite tied together: fibrin deposition contributes to catheter colonization by multiple pathogens, as shown previously by this group, and one of these pathogens in particular, *E. faecalis*, produces a protease that enhances colonization for this organism via

manipulation of fibrinolytic pathways. *E. coli* and *Candida* were presented without similar context (Figs. 5, 6). Do these organisms also produce proteases that function similarly to SprE? Do they have other colonization advantages, so they don't need to degrade Pm/Pg to establish UTI? Additional discussion, perhaps tied in with the model in Extended Data Fig. 5, would help connect these stories.

Other comments follow.

1. Lines 29 and 57-8, multi-drug resistance. CAUTIs are a problem for many reasons beyond antimicrobial resistance, as noted further below in the discussion about biofilms, and thus, these statements are an oversimplification. AMR is more of a concern on the horizon. CAUTIs are currently difficult to treat because they are polymicrobial, the catheters are constantly colonized by external and fecal microbiota, and distinguishing inevitable bacteriuria from infection is challenging in many patient populations, especially when a catheter is in place.
2. Line 38: Spell out "Enterococcus" at first mention.
3. Line 100: add citation
4. Line 109, Fig 1f, and Fig 6f/line 966: organ weight is being used as a proxy for edema, but there can be other contributors to organ weight. This seems especially apparent in 7-14d histopathology images in Fig 1g. Likewise, organ weight is used to make a statement about significant exacerbation of bladder inflammation on line 116. These sections need to be clarified, either by using blinded scoring of sections to quantify edema and other markers of inflammation/cystitis (preferred), or changing the line 109 observation, Fig 1, etc, to state organ weight is increasing at 12h and speculating that this is predominantly due to edema. Please point out relevant features on the sections in Fig 1g.
5. Lines 152, Tables S1 and S2: the reference to Fig 2b should point to 2c as currently written, although Fig. 2 would be clearer if b and c were switched so the \pm catheter data were together and the infection data afterward. Also, this line states that Fg is among the most abundant proteins and points to Fig 2b, but overall abundance relative to other proteins is not shown in this Fig. Please give a relative indication in the text and also refer to Table S1. In addition for Table S1, please add footnotes to explain the colors in this table. Presumably they correlate with the colors in Fig 2 but it's not stated. Related, line 162: this paragraph is about the patient proteomics data and should only refer to Table S2, not S1 (mouse). In Table S2, correct the typo "peptide" at the top.
6. Line 189: why was dissemination detected in the experiments in Fig 1 but not in Fig 3e,f, even for +/- mice? How variable is dissemination? Can a statement about the role of the mutant backgrounds be made for spleens and hearts when the wild-type comparator didn't disseminate in Fig 3? Please provide context for the different results. Also, in Fig 3, please place panels b-f and g-k in the same organ order to facilitate comparison.
7. Lines 270-1: delete or modify the part of the sentence after the first reference to Fig. 5. A direct comparison can't be made that the mutant had higher colonization than wild type in a separate experiment, especially given that the values seem very close (between 10^7 to 10^8 CFU in both cases). What is wt colonization in Pg -/- or PGB backgrounds?
8. Lines 311, 331, 960: provide citation and qualification for the three main uropathogens. For line 331, consider "by three of the most prevalent pathogens" and for line 960, "of three CAUTI pathogens." The prevalence of given pathogens in CAUTI studies famously changes depending on variables including cultivation methods, inclusion of polymicrobial infections, duration of catheterization, and patient population. Other lists of "top three" pathogens include *Klebsiella*, *Pseudomonas*, and *Proteus* (e.g., PMID: 24484578, 27858954).

9. Line 364: add "During catheterization, ..."
10. Line 374: overly broad statement. In the cited study, it was noted that *E. faecalis* was rapidly cleared from murine bladders when a catheter was not implanted; is this truly an example of uncomplicated UTI? Other publications using *E. coli* (PMID: 15972487) or examining human populations (PMID: 18040727, PMID: 9841836) make a stronger case for correlating UTI severity with IL-6.
11. Line 388: cite "pioneer species"
12. Line 473: *Candida* is missing from Table S6. More importantly, details for distinguishing between the 3 species in the polymicrobial experiment seem to be missing from either the methods or the results.
13. Line 544: specify the molecular weight cutoff that was used.
14. Line 553: define "hpc."
15. Fig 2b: what is the unit associated with these numbers? Was a statistical test applied? It's not apparent from this table that Fg is higher in infected samples compared with catheter-only.
16. Line 949: bacterial burdens were measured...
17. Fig. 6h: What are the last three lanes, with no actin control and no correlate with the densitometry in 6i?
18. On all figure axes, when the label is "log," the numbering should be integers instead of exponents.

Typos:

Line 57: regimen

Lines 289 and 350: should point to Table S5, not S3.

Lines 380 and 886: clot (not cloth)

Lines 592-3 reference Fig. 7, which is nonexistent. Presumably this should be Fig. 6.

Reviewer #3 (Remarks to the Author):

Overall this is an interesting body of work expanding the role of Fg/fibrin deposition on the development of CAUTI and possible translation into the identification of higher risk patients. There are several questions that arise:

1) The authors state that IL-1a, IL-1b, IL-6 and TNF-a levels were increased in urine from catheterized patients, however this was compared to non-catheterized controls. Considering that the catheterized individuals did not serve as their own controls (i.e. the urine cytokines post catheterization was not compared to pre-catheterization levels in the same individuals), how can the authors be sure that the elevated cytokine levels were the result of catheterization rather than a different underlying condition?

2) The authors state that infection increases Fg production and accumulation on the catheter surface. Have the authors studied the distribution of bacteria within the fibrinogen? Would released fibrinogen not potentially inhibit bacterial interaction with the catheter surface as the free fibrinogen would bind corresponding adhesions on planktonic bacteria?

3) Did the authors study the proteins found on catheters of animals deficient in fibrinogen to see whether in the absence of fibrinogen other proteins deposit on the surface of catheters? What was the level of bacterial colonization on these catheters? They state that overall bacterial burden was reduced,

however they do not seem to state what that burden was. In addition, do the authors know whether the remaining bacterial burden (albeit lower) may be enough over a longer period? While the overall bacterial load may be lower the development of infection may be delayed. Given that the authors only looked at numbers 1 dpi they may not be able to detect this delay. Did these mice show symptoms of infection? If so then the statement that fibrinogen is "critical" for infection is quite strong.

4) The authors mention that "leveraging these results to develop improved strategies for at-risk patient identification and to inform catheterization guideline, will improve patient QOL and minimize risk for complications." While they mention this it is unclear how the results will do this. How does what has been found here translate to identifying patients at greater risk for CAUTI development? It is already known that those with conditions such as those outlined are at greater risk due to having an overall weaker immune response. While the work described here does present a specific mechanism, these patients would already be considered to be at greater risk.

While the authors do show a role for Fg/fibrin in CAUTI pathogenesis, especially involving *E. faecalis*, the majority of results are from the mouse model. Validation in human samples is restricted to urine from patients catheterized for 24 hrs only and non-catheterized controls. While some inferences can be made from this comparison, a more appropriate and more convincing comparison would have been studying urine from patients pre- and post catheterization as that would have controlled for pre-existing Fg/fibrin levels due to the underlying condition. Furthermore, what happens in patients at later stages? CAUTI is not a significant issue in individuals who are catheterized for 24 hrs but rather for those who have catheters indwelling for multiple days.

We thank the Reviewers for their supportive and enthusiastic response and we appreciate their contribution in improving the clarity of this manuscript.

Reviewer #1:

UTI is one of the most common infections worldwide, and catheter-associated UTI (CAUTI) is more complicated and often lethal due to blood stream infection (e.g., sepsis) and multidrug resistance. The manuscript by Molina *et al.* reported a mechanistic study of CAUTI. The authors established several mouse models with varied genetic deficiencies (e.g., Fg^{-/-}, Pg^{-/-}, tPa^{-/-}) that may enhance or decrease CAUTI; they also developed bacterial cell lines with different mutants (e.g., *E. faecalis* Δ gelE Δ sprE) and thus varied virulence; meanwhile, they performed a number of *in vivo* and histological assays to examine the bacterial growth, biofilm formation, and infection persistence under CAUTI condition. In addition, the authors also reported an interesting connection of tranexamic acid (TXA) with CAUTI, and suggested an unexpected side effect of this common bleeding-control drug. The study is comprehensive and makes biological science in terms of experimental design, controls, and data interpretation. The reviewer is going to only comment on the contents that he has relevant expertise.

Response: *We thank the reviewer for his/her supportive and enthusiastic response and for helping us to improve the clarity of this manuscript.*

1. One of the major concerns is about the novelty of some of the findings. The authors should highlight the findings if they are indeed new, or justify their findings if they are known already.

Response: *In this study, we not only aimed to understand the effects of Fg in a temporal study but also if in absence of Fg could other host proteins promote similar colonization level colonization. Importantly, this study also dissects whether the Fg form (soluble or polymerized into fibrin) impacts CAUTI and subsequent systemic dissemination, which was completely unknown in the catheterized bladder.*

For instance, fibrinogen (Fg) has been known to be a major component of urinary catheter proteome, based on the previous studies from others (PMID: 34249966) and this lab (PMID: 35348114). It is not clear the paragraph (lines 145-155) presented here is just a fact or a new finding of this study.

Response: *We thank the Reviewer for bringing our lack of clarity to distinguish between previous knowledge and our novel findings for this paper. Previous findings have indeed revealed Fg as a major component of the urinary catheter proteome. The main point of this paragraph (**now lines 150-163**) is that *E. faecalis* infection further increased the Fg accumulation on the catheters. We have added a new analysis showing that *E. faecalis*' increase in Fg peptides on catheters is continuous through catheterization time. This graph has now been added as a **Supplementary Figure 1** and is described what are now **lines 159-161**.*

Following the Reviewer recommendation. **In now lines 150-152**, we have acknowledged and cited information that has already been published regarding the Fg as a component of the urine catheter proteome.

Now it reads: "Previously, clinical and animal studies from our and other groups have shown that Fg is a major component of proteins deposited on urinary catheters".

2. This lack of clarity is also true to a few other result paragraphs. *E. faecalis* is known to be one of the most prevalent uropathogens in catheter, and its association with Fg is known as well. The authors should describe clearly whether their data are plain facts or new findings.

Response: We thank the Reviewer for bringing to our attention. We have described this more carefully to distinguish our findings from previous knowledge as the Reviewer suggested **on line 78-79**.

The paragraph now reads as: "we found that *E. faecalis*, a prevalent CAUTI pathogen known to bind to Fg".

3. Lines 152-153 and Figure 2b: The figure caption is not clear to the readers. What the numbers are and how they were calculated.

Response: We apologize for the confusion and thank the Reviewer for pointing out this deficiency.

We changed "higher levels" to "Higher Fg- α , - β , - γ total peptide counts" **in lines now 161-162 and Figure 2b**. This represents the total number of peptides identified, the peptide counts are available in **Table S1**.

Reviewer #2 (Remarks to the Author):

This manuscript by Molina *et al.* builds on prior work showing the important contribution of fibrinogen deposition in the urinary tract to catheter-associated UTI. In the current study, one agent of CAUTI, *Enterococcus faecalis*, is shown to use a protease, SprE, to degrade plasmin/plasminogen. Experimental UTI in a variety of mice with mutations in clotting and fibrinolytic systems further demonstrates that fibrin/fibrinogen increases bacterial burden, and this was also true for another bacterial pathogen, *Escherichia coli*, and a fungal pathogen, *Candida albicans*. Finally, the authors report that an antifibrinolytic medication prescribed during surgeries, tranexamic acid, caused a higher bacterial burden during experimental UTI using the murine CAUTI model.

The findings of 1) a bacterial mechanism for manipulating host plasminogen, 2) multifactorial use of mutant mice to assess the contributions of Pg/Fg toward increasing bacterial burden during UTI, and 3) a potential major contributor to CAUTI vulnerability (routine use of anticlotting medication TXA in surgical patients) make this an exciting manuscript. Overall, this was a very interesting study that nevertheless has some major concerns that need to be addressed before publication.

Response: We thank the Reviewer for their insightful and encouraging comments.

Major items:

1. I have several concerns and comments on the comparisons between mouse and human data.

A. line 160: It's interesting that approximately twice as many proteins were identified on mouse catheters as human catheters. Naively, I would expect the much larger surface area available on patient catheters would allow more proteins to be detected. An alternative explanation is that the unanchored catheter fragment in the tiny mouse bladder is much more disruptive than a standard Foley with an inflated balloon (compare the architecture of the naïve bladder in Fig 1g to the catheterized samples). This intriguing and important proteomics comparison will help inform the strengths and limitations of this model. Do the data support this interpretation as an important caveat to understanding the catheter implantation model? These points should also be considered in the conclusion on Line 407.

Response: *The Reviewer makes a good observation. In previous studies in humans and mice, we have found that protein deposition on the catheter surface is stochastic (PMID: 27795399; PMID: 35348114; PMID: 36867691; PMID: 26827873; PMID: 28973850). There are several factors that may affect protein deposition such as: 1) urine volume, which may reduce the interaction of the proteins in the bladder lumen with the catheter surface; 2) the extent of tissue damage, which will affect the recruitment of several proteins to the catheterized bladder; 3) the high abundance of some proteins may determine other protein attachment, for example, high abundance of some proteins may compete for surface binding and may hinder that attachment of other proteins reducing the protein diversity; 4) catheter manipulation, for example, the method of catheter removal may also have a mechanistic impact in how many proteins remain present on catheters.*

Furthermore, catheter removal from the host. For example, catheter removal from the mouse bladder is done carefully, avoiding to disrupt deposited proteins on catheters and put them immediately on the buffer. On the other hand, patient catheter removal requires pulling the catheter through the urethra, which may randomly strip some proteins off.

This is addressed now in the discussion (in lines 431-437): "In our comparative catheter proteomic analysis between human and mouse, we found our human catheters had less abundance of deposited proteins than in mouse catheters. This could be explained by the stochastic nature of protein deposition due to urine flow or by catheter removal from the host. For example, patient catheter removal requires pulling the catheter through the urethra, which may randomly strip some proteins off. Despite this, the shared proteins are involved in the same host pathways, further confirming that....."

B. line 164: Likewise, for translation to the medical setting, the human proteins that did not overlap with mice are also important. Please comment on interesting features of these proteins: are these proteins only produced in humans? Do the differences reflect more severe inflammation in the mouse catheter implantation model vs clinical catheterization?

Response: *We appreciate the insightful comments provided by the Reviewer. Following the Reviewer's advice, we have looked at the proteins that are differentially found between human and mice catheters. We found that those proteins are present in humans and mice. Additionally, we performed network analysis of those proteins and added to the manuscript (Supplementary Figure 2), finding that some of the proteins involved the same immune and hemostasis processes described in the shared proteins (Figure 2). This further supports that similar pathways are induced in the bladders of catheterized*

patients and mice. This may also relate to the previous question of the factors that may impact host protein deposition. As we have mentioned, protein attachment to the catheter is stochastic and other factors may affect the host protein deposition as well as sample acquisition and processing may strip out some of them. Importantly the majority of the most abundant proteins were shared between patients and mice (total proteins and their abundance [Table S1 and S2]). This now is described **in now lines 179-182**.

C. Line 75: Substitute “mice” for “host populations.” Likewise, soften or qualify line 308 “perpetual inflammatory positive-feedback loop.”

Response: We thank the Reviewer for these suggestions. We have now substituted “host population” for “mice with genetic or acquired fibrinolytic-deficiencies” **in line 76**. Now in **line 324**, we have now changed “perpetual inflammatory positive-feedback loop” to “positive-feedback loop between the inflamed bladder and Fg in the catheterized host.”

IL-6 measurement in human urine is interesting but should not be overinterpreted to make a direct connection to *E. faecalis* or fibrinogen without further investigation; for example, degradation of plasminogen in patient urine when *E. faecalis* but not other uropathogens are present.

Response: We thank the Reviewer for this suggestion. **Lines 146-148** was changed to, “IL-6 is could be an important cytokine during human and mouse CAUTIs and, in mice, IL-6 may communicate with the liver’s Fg expression and release, resulting in its accumulation in the catheterized bladder.”

Lines 354, 362: Is there enhanced risk of CAUTI or other UTIs in people with fibrin deficiencies or other bleeding disorders?

Response: This is a great question. Coagulation deficiencies have not been explored in the CAUTI and other UTIs in patients. This further reinforced the major advancement that this study is providing to this field. This study is seminal for future clinical investigations. For example, our animal study that discovered that Fg was recruited to the catheterized bladders and Fg was deposited into urinary catheters serving as a platform for uropathogens’ biofilm formation (PMC4365981) was the basis study for clinical research, finding that Fg indeed was deposited on urinary catheters retrieved from patients (PMC4965327). Importantly, catheters from patients with CAUTI showed that the pathogen was colocalized with deposited Fg (PMC5080383, PMC5642702).

One of the major challenges in management of CAUTI is to predict when patients will develop CAUTIs and which populations are at-risk for bloodstream infections. Our findings have identified that mice with genetic or acquired fibrinolytic-deficiencies are susceptible to severe and persistent CAUTI and systemic dissemination.

Why could our research have implications in human CAUTI? Our mouse CAUTI model has been shown to faithfully recapitulate the pathophysiology of human CAUTI, suggesting that our findings have the potential to be translated information for human CAUTI, allowing further investigation of coagulopathies in clinical retrospective studies.

Therefore, we have used words such as, “may” or “could” to indicate the potential translation to human CAUTI. To follow the Reviewer comments, we will ensure to be careful and not overstep our statements:

In now lines 378-379 - we have changed to “*may* have a higher risk of developing a CAUTI and *could* be more susceptible to secondary bacterial and fungal infection from a CAUTI.”.

2. Statistics. Multiple figure legends state “values represent means \pm SEM” when medians are shown without error bars (e.g., Fig. 4, line 933, or worse, lines 841-2, which confusingly state “Values represent means \pm SEM. Values represent median.”). The use of medians is typical for these nonparametric datasets, but the statistics need to be checked and clarified to address the nonsensical statements about SEM. Second, the same legends that mention SEM also mention a second test, Mann-Whitney U. This makes more sense for the data presented. However, many figs (e.g., Fig 3g-h) show multiple comparisons, and Mann-Whitney U is a pairwise test. Another test such as Kruskal-Wallis with multiple comparison corrections would be more appropriate for these scenarios. Extended Data Fig. 3 shows what appears to be Means \pm SEM (a measure of normal distribution) but again states that values represent medians and the non-parametric Mann-Whitney U test was used to measure significance. What test was actually used?

Response: We thank the Reviewer for alerting us to this oversight. As the Reviewer noticed, the legend description is incorrect. Mann-Whitney U tests were used in this study. Bars do represent medians and manuscript changes have been made to address this comment. Figure legends have been corrected.

For Figure 4, Kruskal-Wallis was used originally and had similar findings as using multiple Mann-Whitney U tests to determine their statistical significance. However, we proceeded with Mann-Whitneys as opposed to Kruskal-Wallis because of the assumptions made using the Kruskal-Wallis (same variance based on pooled variance) which may not be true for these different genotypes. For example, we observed and addressed a bimodal distribution with *tPA*^{-/-} dissemination to spleens which disrupted the similar variance assumption. Since different genotypes may produce different phenotypes and variance, despite the similar background, that we may not have fully observed with our replicate number, we proceeded with multiple Mann-Whitney's tests.

3. Specificity of SprE toward plasmin and plasminogen. Strong evidence is provided that SprE degrades these proteins, but the specificity toward these two proteins is overstated. First, thrombin is provided as a comparator, but it is much smaller than the other two proteins. By itself, this is not strong evidence for specificity. Inclusion of additional clotting/fibrinolytic factors, especially of comparable size, would be more indicative of specificity. Even testing non-coagulation cascade proteins of comparable size would be useful. Second, the cropped images in Fig 4 g,j,k are difficult to assess; whole blots/gels should be provided in supplemental data. The mysterious cross-reactive material in Fig 4g is not sufficient to convey this information. Fig. 4j lacks a loading control.

Response: We thank the Reviewer for their suggestions. Following the Reviewer's advice, we have now assessed the degradation of other proteins of comparable size that are involved or not in the coagulation cascade. Factor XII (~80 kDa) and Prekallikrein (~90 kDa) were used as they are involved in the coagulation cascade (**Fig. 3a**). We also used Endoplasmin (*HSP090b1*; expected ~90kDa; observed ~110 kDa), which is not part of the coagulation cascade but was found deposited on both mouse and patient catheters

and had an expected size similar to Pg (Table S1 and S2). In this *in vitro* analysis, we found that there is differential degradation of these proteins by SprE. We found that SprE was important for degradation for Prekallikrein but dispensable for Endoplasmin and FXII (See Extended Data Figure 5 and below). This suggests that this is not just size related but it could be related to specific recognition sites. Understanding SprE's specificity is out of the scope of this study and definitely would be further explored in other studies. The new data is now described in lines 257-265 and discussed in lines 401-406.

Regarding Fig. 4g, homogenate samples were incubated with the PBS, Urine or bacterial proteases and they were probed at the same time to assess degradation of Pg and thrombin as well as detection of β -actin as loading control. We designed it in this way because our PGB bladder homogenate samples are precious due to the PGB mice breeding difficulties. Therefore, the membrane was cut in several pieces based on the size to prove for the specific protein. Thus, we don't have the full blot. Since we observed that cross-reacting material (CRM), we decided to use it as another loading control and degradation control. CRM is common in western blots of complex samples and it is not uncommon to use it as extra loading controls (PMID: 19602141; PMID: 19432796). However, thanks to the Reviewer's suggestion, we have shown SprE differential degradation of host proteins (See Extended Data Figure 5 and below), further strengthening our previous conclusions.

Extended Data Figure 5. SDS-PAGE analysis of the proteolytic activity of *E. faecalis* WT and protease mutants' cell-free supernatants against different purified human proteins: (a) Endoplasmin, Prekallikrein, or Factor XII (FXII) and their corresponding degradation quantification by densitometry (b-d). Values represent the median. The Mann-Whitney U test was used to determine significance; *, $P < 0.05$ was considered statistically significant. The horizontal bar represents the median value.

Regarding Fig. 4k, similar to Fig. 4g, the membranes were cut based on size and probed for Fragment E and β -actin separately. Thus, it is not possible to provide the full images, since we don't have them. In the case of Fig. 4j, this was from an in vitro study using human purified proteins. Therefore, no β -actin was present. The focus of this analysis was to further evaluate plasmin-dependent degradation of fibrin by detecting Fragment E by western blot.

4. There seem to be two different messages that aren't quite tied together: fibrin deposition contributes to catheter colonization by multiple pathogens, as shown previously by this group, and one of these pathogens in particular, *E. faecalis*, produces a protease that enhances colonization for this organism via manipulation of fibrinolytic pathways. *E. coli* and *Candida* were presented without similar context (Figs. 5, 6). Do these organisms also produce proteases that function similarly to SprE? Do they have other colonization advantages, so they don't need to degrade Pm/Pg to establish UTI? Additional discussion, perhaps tied in with the model in Extended Data Fig. 5, would help connect these stories.

Response: *We appreciate the suggestion for additional experimentation. Our focus on enterococcal proteases has been based on a previous study that showed that they are critical for establishment of enterococcal colonization during CAUTI (PMID: 29134108). Currently, our knowledge of E. coli and C. albicans requirements during CAUTI are minimal and no proteases have been correlated. While these experiments that the Reviewer is asking may provide helpful insights. Identification of proteases and activity analysis would result in several other manuscripts by their own, which are not needed for the conclusion of this manuscript. Our current finding that SprE degrades plasminogen and plasmin fully support our fundamental conclusions on the role of enterococcal proteases in E. faecalis CAUTI.*

*We have now added in **lines 401-404** a statement regarding the possible role of proteases by other uropathogens: "Furthermore, it would be interesting to understand if other uropathogens produce secreted proteases with similar SprE activity to target the fibrinolytic system to enhance their colonization".*

Other comments follow.

1. Lines 29 and 57-8, multi-drug resistance. CAUTIs are a problem for many reasons beyond antimicrobial resistance, as noted further below in the discussion about biofilms, and thus, these statements are an oversimplification. AMR is more of a concern on the horizon. CAUTIs are currently difficult to treat because they are polymicrobial, the catheters are constantly colonized by external and fecal microbiota, and distinguishing inevitable bacteriuria from infection is challenging in many patient populations, especially when a catheter is in place.

Response: *We agree with the Reviewer. We have revised the text based on the Reviewer's suggestion.*

Abstract (lines 29-30). *Since we have word limit in the abstract, we have added that CAUTI are difficult to treat "partly due to development of multidrug-resistance from CAUTI-related pathogens".*

In line 56-60: “However, the consistent colonization of catheters by external and fecal microflora, along with polymicrobial infections, pose challenges to management and treatment due to the increasing prevalence of antibiotic-resistant CAUTI pathogens^{6,10,11}. Leading the CDC and WHO to classify CAUTI as a serious threat^{12,13}.”

2. Line 38: Spell out “Enterococcus” at first mention.

Response: Thank you for catching this, it has been spelled out (**line 38**).

3. Line 100: add citation

Response: Citations have been added (now **in line 101**).

4. Line 109, Fig 1f, and Fig 6f/line 966: organ weight is being used as a proxy for edema, but there can be other contributors to organ weight. This seems especially apparent in 7-14d histopathology images in Fig 1g. Likewise, organ weight is used to make a statement about significant exacerbation of bladder inflammation on line 116. These sections need to be clarified, either by using blinded scoring of sections to quantify edema and other markers of inflammation/cystitis (preferred), or changing the line 109 observation, Fig 1, etc, to state organ weight is increasing at 12h and speculating that this is predominantly due to edema. Please point out relevant features on the sections in Fig 1g.

PMC6292575; PMC2930321; PMC6701943; PMC3536162; PMC4457352

Response: It is correct that there are several contributors for organ weight. In the field of UTI and CAUTI, bladder weight is an acceptable method for quantifying bladder inflammation as seen in these publications PMC6292575; PMC2930321; PMC6701943; PMC3536162; PMC4457352.

Following the Reviewer suggestion, we have revised the text.

Line 106: from “edema” to “edema and inflammation”

Line 108-109: from “Bladder histological analysis corroborates the gradual edema progression associated with urinary catheterization” to “Bladder histological analysis corroborates the gradual edema progression and increase of the bladder size associated with urinary catheterization”.

Line 110: from “bladder inflammation” to “bladder weight”

Line 112: from “edema” to “bladder weight”

5. Lines 152, Tables S1 and S2: the reference to Fig 2b should point to 2c as currently written, although Fig. 2 would be clearer if b and c were switched so the ± catheter data were together and the infection data afterward. Also, this line states that Fg is among the most abundant proteins and points to Fig 2b, but overall abundance relative to other proteins is not shown in this Fig. Please give a relative indication in the text and also refer to Table S1. In addition for Table S1, please add footnotes to explain the colors in this table. Presumably they correlate with the colors in Fig 2 but it’s not stated. Related, line 162: this paragraph is about the patient proteomics data and should only refer to Table S2, not S1 (mouse). In Table S2, correct the typo “peptide” at the top.

Response: Thank you for catching this. Indeed, it is Fig. 2c. This has been corrected (now **line 158**).

- citation has now been included **in now lines 152 and 160**.
- We also had added a footnote explaining the colors on the **Table S1** and its correlation with **Fig. 2a (lines 941-942)**.
- Table S2 was referenced now in **Line 169**.
- Typo peptide in Table S2 has been corrected.

6. Line 189: why was dissemination detected in the experiments in Fig 1 but not in Fig 3e,f, even for +/+ mice? How variable is dissemination? Can a statement about the role of the mutant backgrounds be made for spleens and hearts when the wild-type comparator didn't disseminate in Fig 3? Please provide context for the different results. Also, in Fig 3, please place panels b-f and g-k in the same organ order to facilitate comparison.

Response: *The Reviewer brings a great point. Yes, dissemination may vary and appear to be stochastic at times and certainly something to be looked at in the future. However, we have observed that it depends on the colonization of the bladder. If we focus on 24hpi, which is what we continue to use for experiments in Fig. 3, we see that kidney distribution has large variability and may even be bimodal. Furthermore, the dissemination across time is variable as we can observe between timepoints 12 hpi-14 dpi. Therefore, to observe some or little dissemination across different experiments can be expected. Yet, we performed experiments multiple times when comparing different genotypes, so the intra-experiment variability is consistent, and would provide a control for mutated genotypes. To account for this variability during statistical analysis, we performed Mann-Whitney U tests to compare ranks instead of means. Overall, we do observe dissemination in Fig 1c,d,e (24hpi) which are similar to those of WT (+/+) mice and in Fig. 3d,i,j,k, which were also 24hpi.*

We acknowledge and apologize for the misplacement of kidneys and spleen in Fig. 3i,j. This happened during the reorganization of this figure. We acknowledge that this will cause confusion, considering all other graphs follow the order of bladder, catheter, kidneys, spleen, and heart. The order of the graphs has been organized to match in Fig. 3 and be consistent throughout the manuscript.

7. Lines 270-1: delete or modify the part of the sentence after the first reference to Fig. 5. A direct comparison can't be made that the mutant had higher colonization than wild type in a separate experiment, especially given that the values seem very close (between 10^7 to 10^8 CFU in both cases). What is WT colonization in Pg -/- or PGB backgrounds?

Response: *We thank the reviewer for noticing this and how phrasing may lead to some confusion. To note, we were not trying to directly compare data from to graphs to arrive at a conclusion, but instead noticing an increased colonization in Pg-/- and PGB mice occurring once again but with mutant E. faecalis strain. The sentence now **in lines 287-288** now concludes at, "...exhibiting higher than the WT strain colonization"*

Regarding the question, what is wt colonization in Pg -/- or PGB backgrounds? That information is in **Figure 3**.

8. Lines 311, 331, 960: provide citation and qualification for the three main uropathogens. For line 331, consider "by three of the most prevalent pathogens" and for line 960, "of three CAUTI pathogens." The prevalence of given pathogens in CAUTI studies famously changes depending on variables including cultivation methods, inclusion of polymicrobial infections, duration of

catheterization, and patient population. Other lists of “top three” pathogens include Klebsiella, Pseudomonas, and Proteus (e.g., PMID: 24484578, 27858954).

Response: We thank the Reviewer for this insightful suggestion. It is true that pathogen prevalence may differ due to patient population, healthcare facilities, and geographically locations. Our statement is based on the 2016 National Healthcare Safety Network (NHSN) review (PMC6857725). We have improved our statements by adding the proper citations (PMID: 31548790 and PMID: 27573805) in **lines 308-311 and 327**.

- **In lines 310 and 1037-1038.** We have changed “the three most prevalent pathogens” to “by three highly prevalent CAUTI pathogens”
- **Line 327.** It has changed from “the three most prevalent uropathogens” to “with all three uropathogens”

9. Line 364: add “During catheterization, ...”

Response: We thank the Reviewer for bringing this to our attention. It has been added (**line 381**).

10. Line 374: overly broad statement. In the cited study, it was noted that *E. faecalis* was rapidly cleared from murine bladders when a catheter was not implanted; is this truly an example of uncomplicated UTI? Other publications using *E. coli* (PMID: 15972487) or examining human populations (PMID: 18040727, PMID: 9841836) make a stronger case for correlating UTI severity with IL-6.

Response: We thank the Reviewer for improving the manuscript and providing additional reports to support that IL-6 is repressed during uUTI. Indeed, the PMID: 15972487 study found that *E. coli* UTI89 repressed IL-6 production by 5637 and T4 cell lines. Furthermore, in the clinical study (PMID: 9841836) in children, it was found that IL-6 levels were important to differentiate between lower UTI and pyelonephritis, finding that high levels of IL-6 were correlated with pyelonephritis than with lower UTI. Importantly correlation of IL-6 levels with uUTI was found during severe and febrile uUTI (PMID: 9841836). However, that is not the case for less complicated, recurrent, or asymptomatic uUTI (PMID: 25569799). In fact, animal studies have shown that at early UPEC UTI, TNF α upregulation is correlated with recurrent uUTI (PMID: 31429405; PMID: 37037942). Importantly, during CAUTI, TNF α is repressed. All these studies further strengthen our hypothesis that UTI and CAUTI have different pathophysiologies.

We have modified our statement and added the corresponding citations to reflect this discussion in **line 391-395**.

11. Line 388: cite “pioneer species”

Response: We thank the Reviewer for bringing this to our attention. Citation has been added (**Line 414**).

12. Line 473: *Candida* is missing from Table S6. More importantly, details for distinguishing between the 3 species in the polymicrobial experiment seem to be missing from either the methods or the results.

Response: We thank the Reviewer for bringing this to our attention. *Candida* has been added in **Table S6**. Additionally, we have further described the differential selection for the pathogens in **lines 506-512**.

“For differential selection and growth of the pathogens, *Candida albicans* was grown in YPD agar plates containing 500 µg/ml of kanamycin, a concentration that will exclude both *E. faecalis* and *E. coli* growth. *C. albicans* grows very well in YPD media due its acidic pH, and *C. albicans* growth is restricted on LB and BHI media. *E. faecalis* was selected on BHI plates supplemented with 50 µg/ml of rifampicin and fusidic acid; these antibiotic concentrations inhibit *E. coli* growth. *E. faecalis* growth is restrictive in YPD and LB media. Verification of the strains was further confirmed by their distinctive colony morphologies.”

13. Line 544: specify the molecular weight cutoff that was used.

Response: 10k molecular weight cutoff was used. This has been added in now **line 580**.

14. Line 553: define “hpc.”

Response: It has been defined (now **in line 589**).

15. Fig 2b: what is the unit associated with these numbers? Was a statistical test applied? It's not apparent from this table that Fg is higher in infected samples compared with catheter-only.

Response: We thank the reviewer for bringing this to our attention. The units in **Fig. 2b** are total peptide counts of Fg- α , β , γ found on catheters. The heatmap was chosen in this graph to visualize the increase in peptides for multiple samples. We have now also included correlation analysis to better visualize the constant increase in Fg peptides when bladders are catheterized and infected with *E. faecalis*, as opposed to mock-infected. The line graph on the right represents the Fg- α , - β , and - γ peptide counts as function of time post-catheterization (solid line) or -catheterization and infection (dashed line). Using an F test, we demonstrated that although the slopes of lines do not differ from catheterized and mock-infected mice to infected mice, the elevations are significantly different. This suggests that *E. faecalis*' increase in Fg peptides on catheters is continuous through catheterization time. This graph has now been added as a **Supplementary Figure 1** and is described in **lines 159-161**.

Supplementary Fig. 1. *E. faecalis* infection significantly increased Fg accumulation on urinary catheters overtime. Line graph of total Fg- α , - β , and - γ peptide counts found on catheters of mice catheterized and infected

with *E. faecalis* (CI) or catheterized and mock-infected (C) over a temporal study from naïve (0 hrs) to 14 days. Simple linear regressions were conducted to determine the association lines between time and total peptide counts of all three peptide counts from C and CI cohorts. Then, an F-test was conducted to quantify the difference between lines from C and CI when analyzing Fg- α , Fg- β , and Fg- γ peptide counts: Fg- α ($F=5.039$, $p=0.0403$); Fg- β ($F=5.110$, $p=0.0391$); Fg- γ ($F=6.386$, $p=0.0232$). *, $P < 0.05$ was considered statistically significant.

16. Line 949: bacterial burdens were measured...

Response: We thank the Reviewer for alerting us to this oversight. This has been changed as the Reviewer suggested (now in **lines 966-967 and lines 1026-1027**).

17. Fig. 6h: What are the last three lanes, with no actin control and no correlate with the densitometry in 6i?

Response: We thank the Reviewer for pointing out the lack of clarity. In response, we have better explained the experiment in Figure 6h. To answer the reviewer question: the last three lines are an *in vitro* analysis to understand the ability of TXA to inhibit plasmin activity against fibrin using purified proteins. Therefore, B-actin won't be present in this *in vitro* assay. Plasmin activity was measured by production of Fragment E, which was analyzed by western blot. The first line is fibrin by itself, no Fragment E was detected. Second line, plasmin was added to fibrin, observing the degradation of fibrin by detecting Fragment E. Third line, fibrin was incubated with plasmin and TXA, reducing production of Fragment E, indicating that TXA inhibited plasmin activity against fibrin.

To improve the clarity, the western blot has been separated into two panels homogenized bladders (**Fig. 6h**) and *in vitro* validation (**Fig. 6j**). The data are further described in the result section (**lines 317-320**), methodology (**lines 629-630**), and Figure 6 legend (**lines 1048-1050**).

18. On all figure axes, when the label is "log," the numbering should be integers instead of exponents.

Response: We appreciate the Reviewer suggestion. We have modified the y-axis title to bacteria CFU/organ or catheter or fungal CFU/organ or catheter.

Typos:

Line 57: regimen.

Response: It has been changed (**line 56**).

Lines 289 and 350: should point to Table S5, not S3.

Response: Thank you for catching that. Change has been made (**lines 304 and 367**).

Lines 380 and 886: clot (not cloth).

Response: *It has been changed (now lines 401 and 965).*

Lines 592-3 reference Fig. 7, which is nonexistent. Presumably this should be Fig. 6.

Response: *We thank the Reviewer for alerting us to this oversight. The reviewer is correct, these lines should refer to Figure 6. These have been modified (lines 629-630).*

Reviewer #3 (Remarks to the Author):

Overall this is an interesting body of work expanding the role of Fg/fibrin deposition on the development of CAUTI and possible translation into the identification of higher risk patients. There are several questions that arise:

1. The authors state that IL-1a, IL-1b, IL-6 and TNF-a levels were increased in urine from catheterized patients, however this was compared to non-catheterized controls. Considering that the catheterized individuals did not serve as their own controls (i.e. the urine cytokines post catheterization was not compared to pre-catheterization levels in the same individuals), how can the authors be sure that the elevated cytokine levels were the result of catheterization rather than a different underlying condition?

Response: *The Reviewer brings a good point and we agree that it is not known whether the patients have underlying conditions, or comorbidities, that may promote IL-6 or other cytokines and acknowledge that this is a limitation of this particular assay. Unfortunately, no urine samples were collected before catheterization. Definitely, in future clinical studies urine pre and post catheterization should be collected.*

The limitation of the analysis has been added in lines 144-145 and we have softened our conclusion in lines 146-148.

2. The authors state that infection increases Fg production and accumulation on the catheter surface. Have the authors studied the distribution of bacteria within the fibrinogen? Would released fibrinogen not potentially inhibit bacterial interaction with the catheter surface as the free fibrinogen would bind corresponding adhesions on planktonic bacteria?

Response: *The Reviewer asks a great question. In previous studies (PMID: 35348114; PMID: 36867691; PMID: 26827873; PMID: 33066191; PMID: 31235751; PMID: 28973850; PMID: 25232179), we have found that uropathogen prefers to bind to Fg on the catheter than to the catheter by itself. Furthermore, immunostaining studies (PMID: 35348114; PMID: 36867691; PMID: 33066191; PMID: 31235751; PMID: 28973850; PMID: 25232179) have shown that uropathogen is associated with Fg and fibrin in the bladder lumen and urothelium. So, instead of fibrinogen inhibiting bacterial interaction with catheters directly, fibrinogen actually sticks on to catheters to form a scaffold for bacteria to bind to and colonize. As the Reviewer mentions, free fibrinogen may act as decoy molecule reducing binding to the catheter, which is in fact what we believed is happening in the Fg^{AEK} mice, which only has soluble or free Fg. We believe that Fg may act as decoy receptors to prevent binding to urinary tract surfaces, resulting in the pathogen's expulsion by urine flow. We have previously discussed this possible phenomenon in lines 423-426. However, when fibrin fibers are available in the catheterized bladder, fibrin may serve as a platform for free Fg-pathogen complex attachment, between micturation events. This may result in the free Fg-pathogen*

complex to get trapped within the fibrin fibers. Our study also showed that excessive clotting/fibrin accumulation ($Pg^{-/-}$ mice and TXA experiments; **Fig. 3 and 6**) increases in colonization. So, compared to soluble fibrinogen, clotted fibrin promotes pathogen adhesion, colonization, and persistence as this manuscript details.

3. Did the authors study the proteins found on catheters of animals deficient in fibrinogen to see whether in the absence of fibrinogen other proteins deposit on the surface of catheters? What was the level of bacterial colonization on these catheters? They state that overall bacterial burden was reduced, however they do not seem to state what that burden was. In addition, do the authors know whether the remaining bacterial burden (albeit lower) may be enough over a longer period? While the overall bacterial load may be lower the development of infection may be delayed. Given that the authors only looked at numbers 1 dpi they may not be able to detect this delay. Did these mice show symptoms of infection? If so then the statement that fibrinogen is "critical" for infection is quite strong.

Response: We thank the Reviewer for this interesting question. Unfortunately, we did not study the proteome of catheters from $Fg^{-/-}$ mice. However, we showed that the bacterial burden on catheters from $Fg^{-/-}$ mice after 24hpi (**Figure 3c**) are significantly reduced. Unfortunately, the temporal infection analysis cannot be done in $Fg^{-/-}$ because they died due to the excessive bleeding caused by the catheter-induced damage.

In the future, we could evaluate temporal colonization in other mice where the bleeding disorder is mild. As we mentioned, when Fg is absent or soluble bacterial burden decreases. Conversely, mice without the ability to remove polymerized fibrinogen ($Pg^{-/-}$ or TXA) had higher colonization. Our previous studies (PMID: 35348114; PMID: 36867691; PMID: 33066191; PMID: 31235751; PMID: 28973850; PMID: 25232179; PMID: 27795399) have shown that interaction with Fg in the catheterized bladder is essential for persistence, blocking Fg -pathogen interaction via vaccination, antibody neutralization, or preventing Fg deposition on the catheter results in defective colonization. Therefore, we suspect that in cases where Fg cannot accumulate or is not present ($Fg^{-/-}$), we do not expect pathogen burden would increase over time.

4. The authors mention that "leveraging these results to develop improved strategies for at-risk patient identification and to inform catheterization guideline, will improve patient QOL and minimize risk for complications." While they mention this it is unclear how the results will do this. How does what has been found here translate to identifying patients at greater risk for CAUTI development? It is already known that those with conditions such as those outlined are at greater risk due to having an overall weaker immune response. While the work described here does present a specific mechanism, these patients would already be considered to be at greater risk.

Response: We appreciate the Reviewer's insightful comment. It is true that patients with weaker immune responses could be more at risk to develop a CAUTI.

Currently, a major challenge is to predict when patients will develop CAUTIs and which populations are at-risk for bloodstream infections. This is crucial since 25% of sepsis cases are from urinary isolates. Furthermore, prior to this study, no known patient population was identified to be more likely to develop severe CAUTIs, thus prior patient information regarding fibrinolytic deficiencies would not be known to worsen CAUTI outcomes.

In the last decade, we have been able to corroborate our finding using our mouse model of CAUTI with clinical studies, finding the Fg is a scaffold for biofilm formation by uropathogens (PMID: 35348114; PMID: 36867691; PMID: 33066191; PMID: 31235751; PMID: 28973850; PMID: 25232179; PMID: 27795399). Our findings in this study will help

us to perform retrospective clinical studies to understand whether coagulopathies or other factors could predispose the patient to CAUTI and urosepsis.

Importantly, we don't know whether coagulopathies may affect the immune response. Currently, we are starting to investigate whether the immune response is compromised during fibrinolytic deficiencies.

While the authors do show a role for Fg/fibrin in CAUTI pathogenesis, especially involving *E. faecalis*, the majority of results are from the mouse model. Validation in human samples is restricted to urine from patients catheterized for 24 hrs only and non-catheterized controls. While some inferences can be made from this comparison, a more appropriate and more convincing comparison would have been studying urine from patients pre- and post catheterization as that would have controlled for pre-existing Fg/fibrin levels due to the underlying condition. Furthermore, what happens in patients at later stages? CAUTI is not a significant issue in individuals who are catheterized for 24 hrs but rather for those who have catheters indwelling for multiple days.

Response: *We certainly agree with the Reviewer's comments and acknowledge the limitations of inferring patient results from our mouse model results as well as the limitations from not performing pairwise comparisons on patient urine samples pre- and post-catheterization to rule out the effects of any comorbidities. The limitation of the analysis has been added in lines 144-145. We plan to further investigate this in future studies.*

REVIEWER COMMENTS

Reviewer #2 (Remarks to the Author):

Thank you to the authors for their thorough and conscientious response. Most of my concerns have been sufficiently addressed, except one:

Original comment:

many figs (e.g., Fig 3g-h) show multiple comparisons, and Mann-Whitney U is a pairwise test. Another test such as Kruskal-Wallis with multiple comparison corrections would be more appropriate for these scenarios.

Author response:

For Figure 4, Kruskal-Wallis was used originally and had similar findings as using multiple Mann-Whitney U tests to determine their statistical significance. However, we proceeded with Mann-Whitneys as opposed to Kruskal-Wallis because of the assumptions made using the Kruskal-Wallis (same variance based on pooled variance) which may not be true for these different genotypes. For example, we observed and addressed a bimodal distribution with tPA-/- dissemination to spleens which disrupted the similar variance assumption. Since different genotypes may produce different phenotypes and variance, despite the similar background, that we may not have fully observed with our replicate number, we proceeded with multiple Mann-Whitney's tests.

Reply:

It may be true that Kruskal-Wallis is not the correct choice of test to incorporate multiple comparisons in the experiments presented here, but a corrective test is required nonetheless (Figs 3, 4, and 5). I recommend a consultation with a biostatistician to choose the most appropriate test.

Reviewer #3 (Remarks to the Author):

Thank you for addressing the concerns of this reviewer. While most have been addressed, I believe that the changes pertaining to the limitation of not having pre-catheterized urines to compare catheterized urines do not adequately explain the limitation.

The authors have added the following statement: "A limitation of this study is that urine samples were not collected from catheterized patients before catheterization as a further comparison. This suggest that IL-6 is could be an important cytokine during human and mouse CAUTIs and, in mice, IL-6 may communicate with the liver's Fg expression and release, resulting in its accumulation in the catheterized bladder."

Merely acknowledging that a limitation exists without an actual explanation of the limitation does not suffice. If the comparison of pre-catheterization vs post-catheterization urine samples was available, the conclusions drawn would have been more convincing. As is, the authors can only make speculations pertaining to IL-6 and its role. So overall, there are two (potentially major) flaws in this: 1) the lack of comparison of pre-to post urine samples and 2) the absence of information pertaining to co-morbidities

that may result in elevated IL-6 levels. The availability of pre-catheterization urines would have answered the existence of co-morbidities that elevate IL-6 and their potential impact on the results. These require specific discussion, as they impact the interpretation of the data and the specific role of IL-6.

We thank the Reviewers for their supportive and enthusiastic response and we appreciate their contribution in improving the clarity of this manuscript.

Reviewer #2 (Remarks to the Author):

Reviewer Reply:

Thank you to the authors for their thorough and conscientious response. Most of my concerns have been sufficiently addressed, except one:

It may be true that Kruskal-Wallis is not the correct choice of test to incorporate multiple comparisons in the experiments presented here, but a corrective test is required nonetheless (Figs 3, 4, and 5). I recommend a consultation with a biostatistician to choose the most appropriate test.

***Response:** We thank the reviewer for suggestions. Even though Mann Whitney test in our statistical analyses is valid and is the standard statistical test for assessing difference in microbial colonization (CFUs) in mouse models of UTI and CAUTI, we followed the Reviewer advice and performed multiple testing correction for the **Figures 3, 4 and 5** to measure significance between sample groups. The new statistical analyses further support our conclusions and now the manuscript includes the statistical analyses and their following description in the statistics paragraph (methodology section):*

***Lines 640-648:** We performed Kolmogorov-Smirnov analytical tests and quantile-quantile (QQ) plot visual normality tests to assess data distributions, and based on these, determined to use appropriate parametric or non-parametric tests. Normally distributed data was tested for difference between two groups with a student T-test, while an ANOVA was used to determine statistical differences between more than two groups. For non-normal distributions, a Mann-Whitney U test was used to determine statistical difference between two groups, while a Kruskal-Wallis test was used to determine significance between more than two groups. Kruskal-Wallis test was performed in **Fig. 3** and **5**. For **Fig. 4**, ANOVA test was used. Statistical tests used are indicated in figure legends.*

*Statistical tests used are indicated in figure legends **in lines 973-974** (Figure 3), **1016-1017** (Figure 4), and **1035-1036** (Figure 5).*

New Figures are and figure legends below.

Figure 3. Impairment of the fibrinolysis enhances enterococcal colonization and systemic dissemination. (a) Coagulation cascade diagram (color boxes correlates with mouse strains used in this study). C57BL/6 wild type (WT) mice and transgenic coagulation mutants in C57BL/6-background looking at clot formation pathway (b-f) or fibrinolytic system (g-k) were catheterized and infected with $\sim 2 \times 10^7$ CFU of *E. faecalis* OG1RF. After 24 hpi, bacterial burdens were measured in bladder tissues (a,g), catheters (b,h), kidneys (c,i), spleen (d,j), and hearts (f,k). The Kruskal-Wallis test followed by a Dunn's test was used to determine significance; *, $P < 0.05$ was considered statistically significant. **, $P < 0.005$; ***, $P < 0.0005$; ****, $P < 0.0001$. The horizontal bar represents the median value. The horizontal broken line represents the limit of detection of viable bacteria. LOD; limit of detection. For CFU enumeration, infections were done in at least three independent experiments with $n = 3-6$ mice depending on the genotype, and data are shown as the bacterial CFU/organ or catheter. Animals that lost the catheter were not included in this work.

Figure 4. SprE, an *E. faecalis* secreted protease, selectively degrades plasminogen and plasmin, inactivating plasmin proteolytic activity against fibrin. (a-c) SDS-PAGE analysis of the proteolytic activity of *E. faecalis* WT and protease mutants' cell-free supernatants against purified (a) plasminogen, (b) plasmin, or (c) thrombin and their corresponding degradation quantification by densitometry (d-f). (g) 24 hrs catheterized PGB bladder homogenates were incubated with *E. faecalis* WT and protease mutants' cell-free supernatants and proteolytic activity against plasminogen and thrombin was monitored by Western blots. (h-i) Densitometry analysis of the Pg and thrombin degradation by bacterial supernatants performed in (g). (j-m) SprE degradation of plasmin results in inhibition of fibrinolysis. To test this, supernatants *E. faecalis* grown in urine were filtered and concentrated, then incubated plasmin was for 4 hrs at 37°C; then each mixture was incubated with purified (j) fibrin or (k) P $g^{-/-}$ mouse bladder homogenates from 24 hrs catheterized non-infected mice. Degradation of fibrin was monitored by detection of Fragment E in incubation with fibrin or bladder homogenates by SDS-PAGE or

western blot analysis, respectively (**j, k**). Fragment E quantification by densitometry obtained in (**l, m**). CRM, cross reactive material. β -actin was used as loading and normalization control. An ANOVA followed by a Tukey's post-hoc was used to determine significance; *, $P < 0.05$ was considered statistically significant. **, $P < 0.005$; ***, $P < 0.0005$. The horizontal bar represents the median value.

Figure 5. Host fibrinolytic deficiency rescued *E. faecalis* Δ gelE Δ sprE colonization deficiency, enhance colonization of uropathogen *E. coli* UTI89, promoting systemic dissemination. (a-e) C57BL/6 WT and coagulation deficient mice were infected with $\sim 2 \times 10^7$ CFU of *E. faecalis* OG1RF Δ gelE Δ sprE or (f-j) uropathogenic *E. coli* UTI89. After 24 hpi, bacterial burdens were measured in bladder tissues (a,f), catheters (b,g), kidneys (c,h), spleen (d,i), and hearts (e,j). The Kruskal-Wallis test followed by a Dunn's test was used to determine significance; *, $P < 0.05$ was considered statistically significant. **, $P < 0.005$; ***, $P < 0.0005$; ****, $P < 0.0001$. The horizontal bar represents the median value. The horizontal broken line represents the limit of detection of viable bacteria. LOD; limit of detection. For CFU enumeration, infections were done at least in 3 independent experiments with $n = 3-6$ mice depending on the mouse genotype, and data are shown as the bacterial CFU/organ or catheter. Animals that lost the catheter were not included in this work.

Reviewer #3 (Remarks to the Author):

Reviewer reply. Thank you for addressing the concerns of this reviewer. While most have been addressed, I believe that the changes pertaining to the limitation of not having pre-catheterized urines to compare catheterized urines do not adequately explain the limitation.

Merely acknowledging that a limitation exists without an actual explanation of the limitation does not suffice. If the comparison of pre-catheterization vs post-catheterization urine samples was available, the conclusions drawn would have been more convincing. As is, the authors can only make speculations pertaining to IL-6 and its role. So overall, there are two (potentially major) flaws in this: 1) the lack of comparison of pre-to post urine samples and 2) the absence of information pertaining to co-morbidities that may result in elevated IL-6 levels. The availability of pre-catheterization urines would have answered the existence of co-morbidities that elevate IL-6 and their potential impact on the results. These require specific discussion, as they impact the interpretation of the data and the specific role of IL-6.

Response: *We thank the author for acknowledging potential limitations in the manuscript. We have further discussed these limitations concerning not having pre- and post- catheterization urine from individual patients. However, these limitations are partially addressed by performing appropriate unpaired significance tests as opposed to paired tests. Nonetheless, corrections to the manuscript are provided below.*

In lines 144-150. *A limitation of this study is that urine samples were not collected from catheterized patients before catheterization as a pair-wise further comparison. Without this data, any comorbidity prior to catheterization could also contribute to the cytokines present in patient urine post-catheterization. However, based on this data provided as is, we can speculate that IL-6 could be an important cytokine during human and mouse CAUTIs and, in mice, IL-6 may communicate with the liver's Fg expression and release, resulting in its accumulation in the catheterized bladder. Further work should be done to further assess this.*